# KFERQ-selective protein autophagy in *Caenorhabditis elegans* depends on LMP-1

**María Gallardo-Campos**[1ᵒ], **Alicia N. Minniti**[2ᵒ], **Juan Hormazabal**[3], **Gonzalo Núñez**[3], **Carlos F. Lagos**[4,5], **Tomás Perez-Acle**[5], **Rebeca Aldunate**[1*], **Iván E. Alfaro**[3,5*]

1 Escuela de Biotecnología, Facultad de Ciencias, Universidad Santo Tomás, Santiago, Chile, 2 Facultad de Ciencias Biológicas, Pontificia Universidad Católica de Chile, Santiago, Chile, 3 Instituto de Ciencias e Innovación en Medicina, Facultad de Medicina Clínica Alemana Universidad del Desarrollo, Santiago, Chile, 4 Chemical Biology & Drug Discovery Lab, Escuela de Química y Farmacia, Facultad de Medicina y Ciencias, Universidad San Sebastián, Campus Los Leones, Santiago, Chile, 5 Centro Ciencia & Vida, Fundación Ciencia & Vida, Santiago, Chile

☉ These authors contributed equally to this work.
* ialfaro@udd.cl (IA); raldunate@santotomas.cl (RA)

## Abstract

Mammalian cells exhibit three autophagy mechanisms: macroautophagy, microautophagy (MIA), and chaperone-mediated autophagy (CMA), each employing unique mechanisms for transporting cellular material to the lysosome for degradation. MIA involves the engulfment of proteins via lysosomes/late endosomes through membrane invagination, while CMA directly imports cytosolic proteins into lysosomes, selectively targeting those harboring the KFERQ pentapeptide motif, helped by the chaperone HSC70. Despite the identification of several genetic markers of these pathways, our understanding of the underlying mechanisms, particularly in MIA and CMA, remains limited. To study CMA in vivo we designed a photoactivatable CMA reporter consisting of a plasmid encoding the KFERQ consensus signal for CMA targeting. We generated transgenic *C. elegans* strains with diverse genetic backgrounds to analyze the role of known molecular components of CMA in mammals. Additionally, we conducted an in-silico analysis of the structural interaction between *C. elegans* LMP-1 or LMP-2 proteins with the HSP-1 chaperone. Results: Our study shows a significant alteration in the distribution pattern of the KFERQ reporter in muscle cells upon induction of selective autophagy (CMA or MIA). We found that the reporter localized into lysosomes only during starvation, which abrogated in the absence of LMP-1. This study validates CMA in *C. elegans* and provides the development of a new tool for understanding selective autophagy mechanisms and their potential implications in various organisms.

## Introduction

The Autophagy-lysosomal pathways (ALPs) encompass a diverse array of intricate mechanisms designed to facilitate the controlled degradation of intracellular

**Data availability statement:** data are all contained within the manuscript and/ or Supporting information files, enter the following: All relevant data are available at https://dataverse.harvard.edu/dataset. xhtml?persistentId=doi:10.7910/DVN/DFPZQX.

**Funding:** National Agency for Research and Development (ANID), https://anid.cl/: Financiamiento Basal para Centros Científicos y Tecnológicos de Excelencia #FB210008 to IEA and CFL, FONDECYT#11161056 to IEA, This work was supported by ANID/ FONDEQUIP/EQY230019 to IEA, and FONDECYT Posdoctorado 2021- #3210596 to GN. This research was partially supported by the supercomputing infrastructure of the NLHPC (#CCSS210001), https://www.nlhpc.cl/. Some strains were provided by the CGC https:// cgc.umn.edu/, which is funded by NIH Office of Research Infrastructure Programs (P40 OD010440). There was no additional external funding received for this study. The funders had no role in study design, data collection and analysis, decision to publish, or preparation of the manuscript.

**Competing interests:** The authors have declared that no competing interests exist.

components by lysosomal catabolic enzymes [1,2]. These pathways play a pivotal role in various physiological processes, including the maintenance of metabolic homeostasis, oversight and turnover of proteins and organelles, modulation of gene expression, regulation of the cell cycle progression, as well as influencing cellular survival and apoptosis, among other functions [1,3]. Notably, the dysregulation of ALPs has been implicated in the pathogenesis and advancement of numerous diseases, encompassing neurodegenerative disorders and cancer [4].

The autophagy pathway can be categorized into three principal types based on the mode by which substrates reach the lysosome [1]. In the context of macroautophagy (MA), intracellular substrates marked for degradation are sequestered within autophagosomes, which are double-membrane vesicles. These autophagosomes subsequently fuse with lysosomes, which deliver the requisite lytic enzymes and enabling acidification essential for the degradation of macromolecules. Depending on the specific physiological context, macroautophagy can proceed non-selectively, as a "bulk" process, or selectively by enlisting autophagosomes to selectively enclose intracellular components, guided by selective autophagy receptors (SARs) [5]. Molecular complexes, protein aggregates, specific organelles or portions of organelles, have been demonstrated to undergo targeted degradation through the process of macroautophagy (MA) in diverse physiological scenarios [6]. Within this framework, microautophagy (MiA) involves the entrapment of protein substrates within membrane invaginations of lysosomes (or endosomes in mammals), resulting in the formation of intraorganellar vesicles that are subsequently degraded along with their cargo. MiA can manifest in a selective or non-selective manner, with the ability to degrade single proteins and organelles (as exemplified by micro-ER-phagy, macropexophagy, macrolipophagy, and micronucleophagy) [7]. The selective or non-selective nature of MiA hinges upon the machinery associated with endosomal sorting complex required for transport (ESCRT) proteins, crucial for the budding and generation of intraluminal vesicles. This intricate process involves different ESCRT complexes, including ESCRT-0 in yeast and ESCRT-III in mammals, protein adaptors, signaling pathways, as well as mechanisms of membrane fission or fusion, which collectively contribute to both selective and non-selective MiA across a spectrum from yeast to mammals [8]. In mammals, a selective pathway of constitutive endosomal MiA has been unraveled, reliant on the HSC70 chaperone for the recognition of specific proteins possessing KFERQ motifs and for interaction with endosomal membrane phospholipids [8].

Meanwhile, chaperone-mediated autophagy (CMA) constitutes a targeted selective lysosomal protein degradation mechanism, characterized by the direct translocation of soluble protein substrates from the cytoplasm into the lysosomal lumen through pores created by multimers of the lysosomal transmembrane protein isoform 2, LAMP2A [1,9]. This distinct pathway, mainly elucidated in mammals, is confined to the degradation of cytosolic proteins harboring peptide sequences biochemically related to the pentapeptide KFERQ. This motif is recognized by the heat shock cognate protein HSC70 and the C-terminal tail of LAMP-2A, positioned at the cytosolic face of lysosomal membranes. Through a mechanism yet to be fully unveiled, involving the formation of multimeric pores by LAMP-2A at the lysosomal membrane and

the collaborative action of cytosolic and luminal HSC70 proteins, CMA substrates undergo unfolding and are translocated to the lysosomal lumen for subsequent degradation [1]. Essential elements in LAMP2A orthologs, including a C-terminal GYXXΦ-like lysosomal targeting sequence, positively charged amino acids within the exposed cytosolic C-terminal domain facilitating interaction with HSC70, and the presence of GxxxG motifs promoting transmembrane helix-to-helix interaction and oligomerization of LAMP2A within membranes, have emerged as critical for effective CMA functionality [10,11].

The LAMP2A-dependent CMA mechanism, similar to the one described in mammals, is conjectured to be exclusive to vertebrates [10,11]. This assumption primarily stems from the fact that LAMP2A, a spliced variant of the LAMP2 gene, is conspicuously absent in yeast, fungi, worms, and other invertebrates [11,12]. In contrast, microautophagy has been identified across diverse organisms spanning from yeast to mammals and is posited to embody an ancestral manifestation of selective autophagy tailored for the degradation of protein substrates harboring the KFERQ motif. *C. elegans* LMP-2 has been suggested as the ortholog of mammalian LAMP2 and as the mediator of CMA in the nematode based in functional assays of glucotoxicity [13]. However, there is no evidence of direct involvement of LMP-2 in chaperone mediated translocation or degradation of protein substrates into lysosomes [14]. In the other hand, some studies have suggested the existence of macroautophagy-independent selective lysosomal degradation pathways of cytosolic proteins in the nematode *C. elegans*, involving lysosomal membrane proteins such as LMP-1 [15–17].

MiA and CMA have been postulated as targets of interest for the treatment of neurodegenerative diseases and cancer [18,19]. Nevertheless, the lack of model systems to study genetic or pharmacological intervention of these pathways has hindered the discovery and validation of therapeutic strategies targeting these pathways. In spite of *C. elegans* being an ideal *in vivo* model system to study and validate therapeutic interventions in different diseases [20], neither MiA nor CMA, or the role of lysosomal membrane proteins (LMPs) in these autophagosome-independent selective autophagy pathways have been characterized in this organism.

In this work, we generated a *C. elegans* transgenic strain that expresses a photoactivatable reporter specific to KFERQ-selective autophagy pathway. Our investigation showed that following the initiation of starvation conditions, this reporter KFERQ protein co-localized with lysosomes as has been observed in other organisms. Moreover, we established that this co-localization event relies on the presence of LMP-1 and HSP-1(HSC70 ortholog), while LMP-2 was found to have no discernible effect in this context. These results, along with our *in silico* modeling of the molecular interactions between LMP-1 or LMP-2 and HSP-1 suggest that, in *C. elegans,* a KFERQ-selective autophagy pathway is dependent on LMP-1 and HSP-1 chaperone.

## Materials and methods

### *C. elegans* strains and culture general methods

Nematodes were cultured at 20°C under standard laboratory conditions according to Stiernagle T., 2006 [21]. The following *C. elegans* strains were kindly provided by the Caenorhabditis Genetics Center (CGC): N2, wild-type strain; PD4482 *lmp-1 (nr2045) X*; and NL5901 *pkIs2386* [*unc-54p*::α-synuclein::YFP + *unc-119*(+)] a transgenic strain expressing YFP-tagged human α-synuclein. The following strains were generated by germline transformation using standard microinjection techniques [21]: ANM77 *pEx[myo-3p*::KFERQ-PAmCherry + pRF54], a muscle-specific expression of the KFERQ peptide fused to the photoactivatable fluorescent reporter PAmCherry. ANM78 *pEx[myo-3p*::PAmCherry + pRF54]; ANM80 *lmp-1(nr2045)X; pEx[myo-3p*::KFERQ-PAmCherry + pRF54].

### Plasmid construction

Plasmid *pmyo3*:: KFERQ-PAmCherry was built from plasmid pPD96.52 *myo-3* (a gift of Dr. Andrew Fire (Addgene plasmid #L2534)), which contains a *myo-3* promoter for muscle expression. A 786 bp fragment containing the KFERQ-PAmCherry sequences were PCR amplified from the plasmid published in [32], using the following primers: forward

5'tcaggaggacccttggctagatgaaggaaactgcagcagcc3' and reverse 5'accggtaccgtcgacgttacttgtacagctc gc3'. We used the IN-FUSION Clone Kit HD (Takara Bio, USA) to clone this fragment into the L2534 plasmid digested with NheI. To generate the p*myo3*::PAmCherry plasmid we double digested plasmid p*myo3*::KFERQ-PAmCherry with BamHI to eliminate the KFERQ sequence.

### *In silico* modeling of LMP-1 and HSP1 interactions

The *C. elegans* HSP-1 protein model was constructed using two bovine (Protein Data Bank accession no. 1YUW [22]; Grimm, to be published; Protein Data Bank accession no. 4FL9) and two human (Protein Data Bank accession number 4KBQ [23]; Protein Data Bank accession number 4PO2 [24]) Hsc-70 crystallographic structures that harbor a 99% in sequence identity with HSP-1. Full HSP-1 structure was modeled with Modeller v10.1 [25] using both bovine structure as a basis for both Nucleotide and Substrate-Binding Domains, and human structures to cover the regions of missing aminoacids in the Substrate-Binding Domain. To construct the LMP-1 and LMP-2 models, the NMR structure of human LAMP2-A ([26]; Protein Data Bank accession no. 4MOF) and LMP structures obtained from AlphaFold Protein Structure Database ( [27]; Accession number Q9GYK0) were used as starting points. Only the transmembrane and cytoplasmic portions were modeled using Modeller. All three new structures were then embedded in a water box with 150 mM NaCl using the CHARMM-GUI web interface [28]. Both *C. elegans* LMP-1 and LMP-2 protein structures were inserted into a 1-palmitoyl-2-oleoylsn-glycero-3-phosphocholine (POPC) lipid membrane. All constructs underwent a 100 ns Molecular Dynamic (MD) simulation using GROMACS package 2021.5 recording 10 steps per ns. Finally, all structures were clustered using the Python's package MDAnalysis [29] and the most populated cluster was selected to continue with docking-based *in silico* protein-protein interaction analysis.

### Protein-protein docking

The initial LMP-1/HSP-1 and LMP-2/HSP-1 complex models were obtained by docking the structures generated previously using ClusPro server [30] with attraction force on the LMP-1 and LMP-2 cytoplasmic residues and a repulsion force on their transmembrane domains. The lowest-energy conformer of the most populated cluster of each complex, which also had to be compatible with the steric hindrance of the membrane, was subjected to a 100 ns MD simulation using the same methods described for LMP-1 and LMP-2. RMSD and RMSF of HSP-1 in each simulation were later calculated using MDA analysis [29].

### RNAi assays

These experiments were performed using standard RNAi feeding protocols [31]. The RNAi HT115 bacterial clones for the LMP-1, LMP-2, LGG-1, LGG2, HSP-1 proteins, correspond to the J. Ahringer collection (*C. elegans* RNAi Collection Ahringer at The Wellcome CRC Institute, University of Cambridge, Cambridge, UK. distributed by Source BioScience https://www.sourcebioscience.com/life-science-research/clones/rnai-34resources/c-elegans-rnai-collection-ahringer/). All clones were donated by the Laboratory of Dr O. Casanueva, Babraham Institute Cambridge England: *lmp-1* (clone C03B1.12), *lmp-2* (Clone C05D9.2), *lgg-1* (Clone C32D5.9), *lgg-2* (ZK593.6), *hsp-1* (F26D1.3). Worms were fed with RNAi from the L1 stage for at least 48h before the starvation trial.

### Analysis of the activity of KFERQ-dependent selective autophagy

To analyze the activity of KFERQ-dependent selective autophagy in *C. elegans*, 20–30 individuals expressing KFERQ-PAmCherry were subjected to photoactivation in 96 well optical plates in M9 buffer using a LED lighting system (LED array cool LED 96, Cetoni, GmBH) for 5–10 minutes at 405 nm with power set to 2.5%. Following photoactivation we induced starvation by subjecting worms to a period of food deprivation (2, 4, or 6 hours) on NGM plates. Presence and

quantification of intracellular KFERQ-PAmCherry fluorescent puncta, indicative of recruitment of the reporter to lysosome organelles [32], was performed by analysis of microscopy images in muscle cells of the nematode head. For microscopy imaging, nematodes were immobilized using 3mM – 5mM levamisole and carefully positioned on 2% agarose agar pads. These samples were analyzed using an BX51 microscope (Olympus, Japan) equipped with a Micropublisher 3.3 RTV digital camera (JH Technologies, USA).

### Colocalization analysis of KFERQ-PAmCherry and lysosomes

For colocalization analysis of the KFERQ-PAmCherry protein and lysosomal markers (SiR700-lysosome, Spirochrome), adult animals transformed with *myo3*::KFERQ-PAmCherry or *myo3*::PAmCherry where incubated with 50 μM of the lysosomal marker in M9 buffer under starvation conditions for 4 hours. Once incubation was completed, individuals were left to recover for 1 hour on NGM plates seeded with OP50, anesthetized with 5mM levamisole, and mounted on PADs agar (2% agarose in water). PAmCherry (ex: 546 nm/ em 572/50) and SiR700-lysosome (ex:635 nm/ em: 647/50) fluorescence were visualized on an Olympus BX81 microscope coupled to a spectral FV1200 laser scanning confocal system using a water immersion 60x (AN1.2) objective at 1024x1024 pixel resolution using Z steps of 300 nm (C.A: 150 μm). Spectral bleed-through between channels and animal autofluorescence was verified with non-transformed and unstained animals. Colocalization analysis of PAmCherry and SiR700-lysosome staining was conducted in 8 bit images of max intensity projections of Z-stacks using the Image J intensity correlation analysis (ICA) plugin software [33].

### Image analysis of KFERQ-PAmCherry puncta and α-Synuclein-YFP aggregates

Image analysis of KFERQ-PAmCherry puncta and α-synuclein-YFP aggregates in transgenic *C. elegans* was performed using ImageJ software. Fluorescent images were acquired from anesthetized worms mounted on agar pads using a 40×air objective on a BX51 fluorescence microscope (Olympus, Japan), maintaining identical exposure parameters across all experimental conditions. Images were converted to 8-bit format, contrast was enhanced, and puncta were quantified using the ImageJ software Cell Counter tool. α-Synuclein-YFP fluorescence aggregates in head muscle cells were also quantified using the same plugin. Phalloidin staining was performed following the standardized protocol described in WormBook (https://wormbook.org Methods in Cell Biology) for visualizing filamentous actin (F-actin) structures in *C. elegans*.

### Thrashing analysis

2-day-old adult worms (strain NL5901) (15–25 animals per experiment) were placed in a 40 μl drop of M9 buffer. After a 2-min recovery period, each individual worm was recorded for 1.0 min and swimming performance was analyzed using the WORMLAB2.0 Software (MBF Bioscience). Turns per minute were automatically examined and quantified as described in Minniti et al 2019 [34].

### Data and statistical analysis

The statistical analyses were performed using the Prism5 software (GraphPad Software Inc.). The specific statistical tests used in each experiment are indicated in the corresponding Fig Legends.

## Results

### Starvation promotes KFERQ-PAmCherry delivery to lysosomes in *C. elegans* muscle cells

In order to demonstrate the presence of selective KFERQ-dependent protein autophagy in *C. elegans* and evaluate the role of this pathway´s components, we generated a transgenic *C. elegans* strain that expresses the photoactivable version of the mCherry fluorescent protein PAmCherry fused to the KFERQ peptide motif (pmyo-3::KFERQ-PAmCherry) in

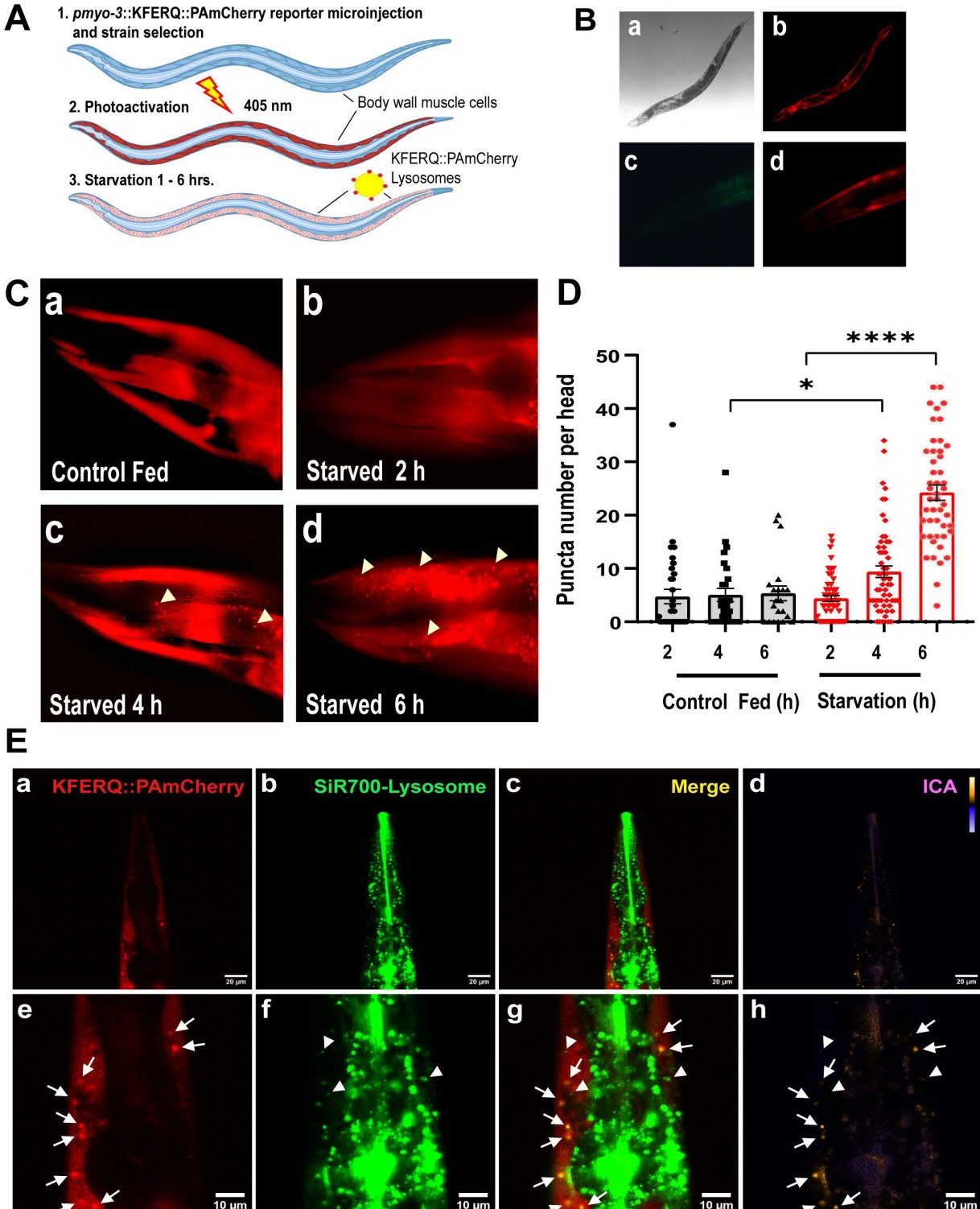

**Fig 1. Starvation increases the localization of KFERQ-PA.mCherry to lysosomes in *C.elegans* muscle cells.** A. The diagram shows our experimental model. The *C. elegans* transgenic strain expresses a photoactivable reporter that contains the KFERQ-PAmCherry sequence. B) Expression of KFERQ-PAmCherry in the muscle cells of a young adult worm upon photoactivation (10X) bright field image of a transgenic worm (a); Fluorescence microscopy image of the same worm (b); Fluorescence images of the worm's head at higher magnification (40X) (c and d). C. Fluorescence images

show a change in the pattern of KFERQ-PAmCherry distribution in the head muscle cells of one-day-old adult worms in response to starvation. An increment in KFERQ-PAmCherry of fluorescent puncta as time of starvation increases (40X). The arrowheads indicate some of the fluorescence puncta. D. The graph shows the quantification of puncta in the head muscle cells in control worms (fed) and in worms that had been starved for 2, 4 and 6 h. Data are mean±SEM from 3 independent assays, between 21 and 58 worms per condition. *p ≤ 0.05, **** p ≤ 0.0001 using Student's t-test. E. Confocal images of the head area of a 1-day-old worm expressing KFERQ-PAmCherry in body wall muscles (red) and stained with the lysosomal marker SiR700-lysosome (green). a) Image of the worm head (scale bar 20 µm) and e) zoomed area (scale bar 10 µm) showing a body wall muscle cell expressing KFERQ-PAmCherry. b) Image of the lysosomes in the same worm head stained with SiR700-lysosome and the corresponding zoomed area f. c) Merge image of a and b. g) Merge image of e and f. d) and h) corresponding pseudocolored PDM images showing clusters high correlation between channels (yellow, white arrows) and low intensity correlation (orange, white arrowheads). Purple color indicates negative inverse intensity correlation.

muscle cells. KFERQ is the substrate peptide for targeting proteins to the selective lysosomal degradation pathway [32] (Fig 1A and B). As expected, the KFERQ-PAmCherry protein is expressed in the cytoplasm of muscle cells where upon photoactivation with UV light shows a diffused florescence distribution (Fig 1B(b) and 1C(a)), which changes progressively into a punctuated pattern when autophagy is activated by subjecting the worms to starvation [32] (Fig 1C(b-d)). As has been described in mammalian cells, KFERQ motif bearing substrates are recognized by HSC70 in the cytosol and delivered to the lysosome where it binds to the LAMP-2A complex, which unfolds the substrates and translocate them for subsequent degradation. In our reporter, mCherry decorates the lysosomes, which shows as a punctuated pattern, before its translocation into the lysosome for degradation (Fig 1A and 1C). Fig 1D, shows the quantification of the number of fluorescent puncta of KFERQ-PAmCherry protein in the head muscle cells of *C. elegans* after photoactivation, under normal growth conditions (Control Fed) and after starvation (Starved) for 2, 4 and 6 hours (see also supporting information S1). We focused our quantification of fluorescence puncta in the head region where individual muscle cells can be clearly identified [35]. Under starvation, the KFERQ-PAmCherry protein progressively changes its distribution to a punctuated pattern and the number of fluorescent puncta become statistically significant after 4 hours of starvation (Fig 1D), suggesting a interaction with lysosomal organelles as has been shown in mammalian cells [32]. To evaluate if upon starvation, KFERQ-PAmCherry colocalizes with lysosomes we stained the transgenic worms with the lysosomal marker SiR700-lysosome, a far-red fluorescent Pepstatin-A peptide. Using confocal microscopy and colocalization analysis, we found a high grade of overlapping between KFERQ-PAmCherry puncta (red) and SiR700-lysosome (green) staining (Fig 1Ed (white arrows) and Supporting information S2). Interestingly, some lysosomal puncta do not colocalize with mCherry fluorescence (Fig 1Eg, white arrowheads), but most mCherry puncta colocalize with the lysosomal marker. We also performed a control experiment in a *C. elegans* strain expressing only PAmCherry protein (were KFERQ sequence is deleted), which shows that PAmCherry localization to lysosomes is lost under starvation in the absence of the KFERQ motif (Supporting information S3). This result indicates that KFERQ-PAmCherry is localized into active lysosomes but that only a fraction of these lysosomes can associate the KFERQ fluorescent protein.

## LMP-1 but not LMP-2 is required for KFERQ-PAmCherry localization to lysosomes under starvation conditions

To test if the molecular components of mammalian CMA are also functional in *C. elegans* we silenced the proposed functional LAMP2A orthologues *lmp-1* and *lmp-2*, and the cytosolic chaperone *hsp-1,* which encodes the *C. elegans* orthologue of HSC70 (See Supporting information S4). We silenced these genes independently in the same strain as described in Fig 1 using RNAi. Fig 2A shows that the silencing of *lmp-1* and *hsp-1* inhibits the increase in the number of KFERQ-PAmCherry puncta after starvation (Fig 2Ac, d and g, h; Fig 2B), suggesting that they have a role in KFERQ selective autophagy. However, the silencing of *lmp-2* does not change the punctuated pattern of KFERQ-PAmCherry after starvation, which is similar to the control (Fig 2A (e, f) and 2B)*,* showing that the LMP-2 protein is not involved in KFERQ-PAmCherry localization with lysosomes.

Therefore, to confirm that LMP-1 is required for CMA in *C. elegans* we performed the same experiments described in Fig 1, but in a transgenic strain that expresses *pmyo-3*::KFERQ-PAmCherry in the *lmp-1(nr2045)* mutant background.

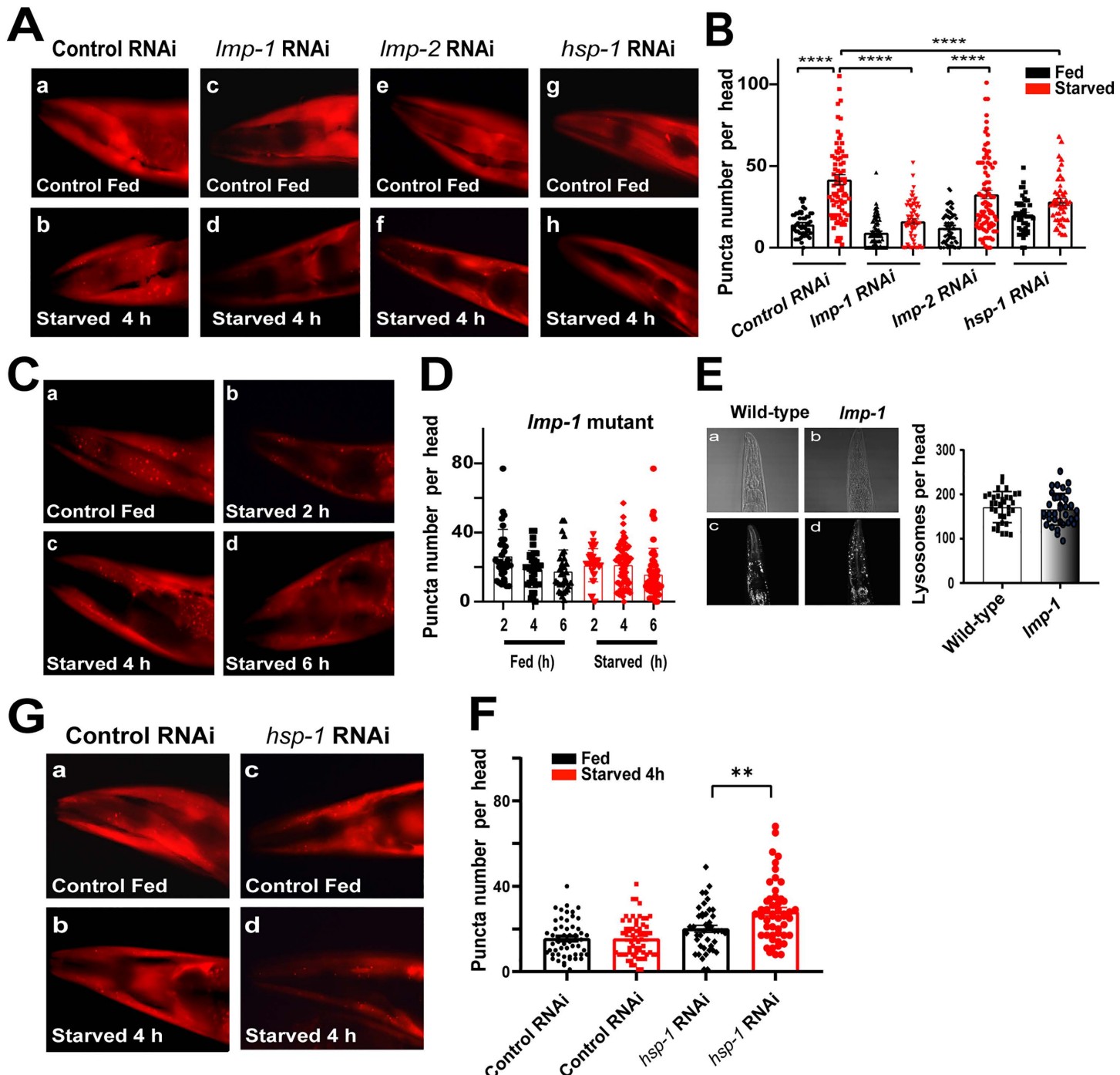

**Fig 2. LMP-1 but not LMP-2 is required for KFERQ-PAmCherry localization to lysosomes under starvation conditions.** A. Representative fluorescence microscopy images showing the head regions of transgenic worms expressing KFERQ-PAmCherry in a wild-type background, under fed conditions and after starvation for 4 h. Control RNAi empty vector (a, b); RNAi targeting *lmp-1* (c, d), *lmp-2* (e, f), and *hsp-1* (g, h) genes (magnification 40X). B. The graph shows quantification of the number of fluorescence puncta per head (data correspond to Mean±SEM of 3 independent assay n ≥ 50 worms per condition **** p ≤ 0.0001 using Student's t-test. C. Fluorescence microscopy images of a worm head region showing the KFERQ-PAmCherry puncta after being photoactivated, in the *lmp-1* mutant strain under normal feeding conditions (a) and after starvation for 2 (b), 4 (c) and 6 (d) hours. D. The graph shows the number of puncta per mutant *lmp-1* worm head under fed conditions and after starvation for 2, 4 and 6 h. Data correspond to Mean ±SEM of 3 independent assays, n ≥ 30 animals per condition. E. Confocal images of a representative worm head showing staining of lysosomes

(SiR700-lysosome). Wild-type (c) and *lmp-1* (d) strains. (a) and (b) are the corresponding bright field images. The graph shows the quantification of lysosomes in at least 30 animals per strain. F. Representative fluorescence microscopy images showing the head region of transgenic worms expressing KFERQ-PAmCherry in a *lmp-1* mutant background. Worms were fed (a) or starved (b) for 4h. Control RNAi (empty vector) (a, b) and RNAi targeting the *hsp-1* gene (c,d) (magnification 40X). G. The graph shows quantification of the number of fluorescence puncta per head (Data correspond to Mean ±SEM of 3 independent assay n ≥ 25 worms per condition ** p ≤ 0.003 using Student's t-test.

This *lmp-1* mutant has a deletion that expands 3 of its 4 exons, suggesting that it is most probably a null mutation. We evaluated KFERQ-PAmCherry localization in these mutant worms under normal feeding and starvation conditions for 2, 4 and 6 hours (Fig 2C and D). We found that in the absence of a functional LMP-1 there is no change in the number of puncta after starvation. Since LMP-1 is an integral lysosomal membrane protein, this observation could be due to a loss of lysosomal function. To rule out this possibility, we quantified the number of functional active lysosomes in this mutant using SiR700-lysosome staining. We found that the *lmp-1* mutant worms have the same number of functional lysosomes as the wild type (Fig 2E) in the absence of LMP-1.

To confirm a role of the HSC70 chaperone function in the KFERQ-selective autophagy pathway in *C. elegans* we evaluated the role of HSP-1 in the localization of KFERQ-PAmCherry- to lysosomes in the *lmp-1* mutant background. In wild-type animals, the knock-down of *hsp-1*, also induces a decrease in the number of KFERQ-PAmCherry puncta under starvation conditions (Fig 2B). In the *lmp-1* knock out mutant background the silencing of *hsp-1* dose not worsens this phenotype (Fig 2F and G), suggesting that these genes may function in the same pathway. However, a small but statistically significant increase in the number of puncta is observed, indicating that might be another pathway involved, independent of CMA, that is activated in its absence.

Next, we wanted to evaluate if the destination of KFERQ-PAmCherry into lysosomes could be also related with macroautophagy. Studies in human cells have previously demonstrated the existence of an interaction between macroautophagy and CMA, whereby cells respond to blockade of one of these pathways by up regulating the other [1]. We found that the downregulation *lgg-2*, a molecular component of the macroautophagy pathway, does not change the pattern of the fluorescent reporter KFERQ-PAmCherry after starvation for 2h, the time period where active macroautophagy in *C. elegans* has been reported [36,37] (Fig 3A and B). However, in the case of *lgg-1*(GABARAP ortholog) we found that there is increase of KFERQ-PAmCherry destination to lysosomes (Fig 3A and B) suggesting that at least during this period a GABARAP-independent selective autophagy might be stimulated by the inhibition of macroautophagy. Additionally, in the

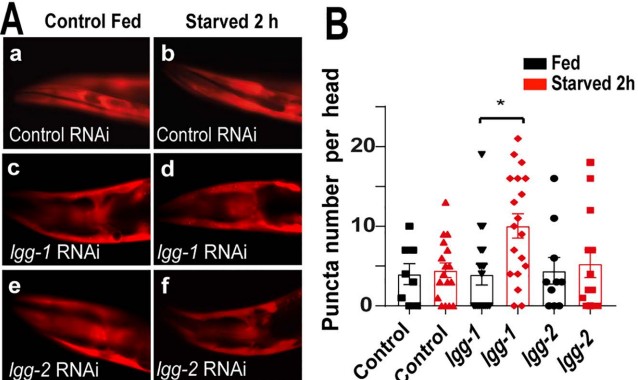

**Fig 3. Downregulation of genes involved in macroautophagy.** A. Fluorescence microscopy images showing the head region of transgenic worms expressing KFERQ-PAmCherry in a wild type background fed and starved for 2h with a control RNAi empty vector (a, b), RNAi targeting the *lgg-1* (c,d) and *lgg-2* (e,f), genes (magnification 40X). B. Quantification of the number of fluorescence puncta per head (Data correspond to Mean±SEM of 3 independent assay n ≥ 10 per condition *p ≤ 0.05, using Student's t-test.

absence of macroautophagy, LMP-1 expression can increase, and its stabilization at the lysosomal membrane could be enhanced to support increased CMA flux [1].

## LMP-1 but not LMP-2 is required for selective autophagy of human α-synuclein, a KFERQ containing substrate

We wanted to test if our results obtained with the KFERQ reporter could be reproducible with a known KFERQ-containing protein. To test if LMP-1 could have a role in the degradation of human α-synuclein in *C. elegans*, we used a strain that expresses this protein in muscle cells (NL5901). α-synuclein is a well-known protein involved in Parkinson's disease that undergoes KFERQ dependent CMA mediated degradation in mammals [38]. This *C. elegans* transgenic strain has been extensively used to evaluate the formation, degradation and toxicity of α-synuclein aggregates, which severely affects worm's motility [39]. Fig 4A compares the actin cytoskeleton (red) of body-wall muscle cells of wild type worms (a) and transgenic worms (c) that express α-synuclein (green). The transgenic worms show the α-synuclein aggregates in muscle cells (Fig 4Ad) as described before [39], and a very disorganized actin cytoskeleton (Fig 4A c), which correlates with severe motility impairments (Fig 4B).

First, we evaluated if long term starvation (5−6 hours), triggers α-synuclein degradation. We placed young adult transgenic worms (α-synuclein aggregates already present) in agar plates without food. Then, we evaluated the worms' motility and quantified the number of α-synuclein aggregates. We found that after starvation there is a 20% improvement in motility (Fig 4C), which correlates with a significant decrease in the number of aggregates (Fig 4D and E). Next, we evaluated the role of LMP-1 and LMP-2 in the decrease of α-synuclein aggregates. Fig 4F and G shows that the knockdown of LMP-1, but not LMP-2, prevents the decrease in the number of α-synuclein aggregates when the worms are subjected to 5 hours of starvation. These results suggest that a specific LMP-1-dependent mechanism participates in a selective long-term autophagic response that induces the degradation of the KFERQ protein α-synuclein in *C. elegans* muscle cells.

## Bioinformatic analysis and molecular modeling suggest that LMP-1, but not LMP-2, is a potential orthologue of LAMP2A in *C. elegans*

Comparison of full-length aminoacidic sequences of LAMP proteins among distinct species suggests the absence of a LAMP2A orthologue and consequently the lack of a CMA-like process in invertebrates. However, some evidence suggest that the *C. elegans* lysosomal proteins LMP-1 and LMP-2 might be orthologues of the human LAMP2 protein [10,11,40,41]. LMP-1 was proposed as *C. elegans* LAMP2 orthologue based on sequence analyses and on the absence of a C-terminal GYXXΦ motif in LMP-2 [42]. On the other hand, *C. elegans* LMP-2 has been suggested as orthologue to mammalian LAMP2 and the mediator of CMA in nematodes based on functional assays of glucotoxicity [14]. However, our *in vivo* functional analyses of *C. elegans* KFERQ-selective autophagy mechanism shows that LMP-1, but not LMP-2 is the best LAMP2A ortholog candidate.

In mammals, LAMP-2A has been shown to be critical for chaperone mediated autophagy activity and regulation, mediating the interaction with HSC70 and CMA substrates to be degraded through the interaction with specific positive residues located in its GXXXXXXG cytosolic C-terminal domain [43]. Additionally, the oligomerization of LAMP-2A enables pore formation mediated in part by intermolecular transmembrane helix-to-helix interactions through GxxxG/GxxxA motifs [26,44,45]. Therefore, we carried out a multiple sequence alignment between mammalian and *C. elegans* C-terminal segments of LAMP proteins considering only the sequences from the luminal hinge-like segment to the cytosolic C-terminal domain. Fig 5A shows that LMP-1 is closer to the LAMP2A cluster than LMP-2 in the average distance tree. Furthermore, *C. elegans* LMP-1 shows higher identity (42%) than LMP-2 (35%) with mammalian LAMP2A C-terminal segment (Fig 5B). Moreover, LMP-1 has a GYXXΦ lysosome-targeting motif in the C-terminal domain positioned at about the same distance from the transmembrane domain as the lysosomal targeting signal of human LAMPs, in contrast to *C. elegans* LMP-2, which has 12 extra amino acids (Fig 5C) [46]. The C-terminal GXXXXXXG cytoplasmic tail of *C. elegans* LMP-1 contains

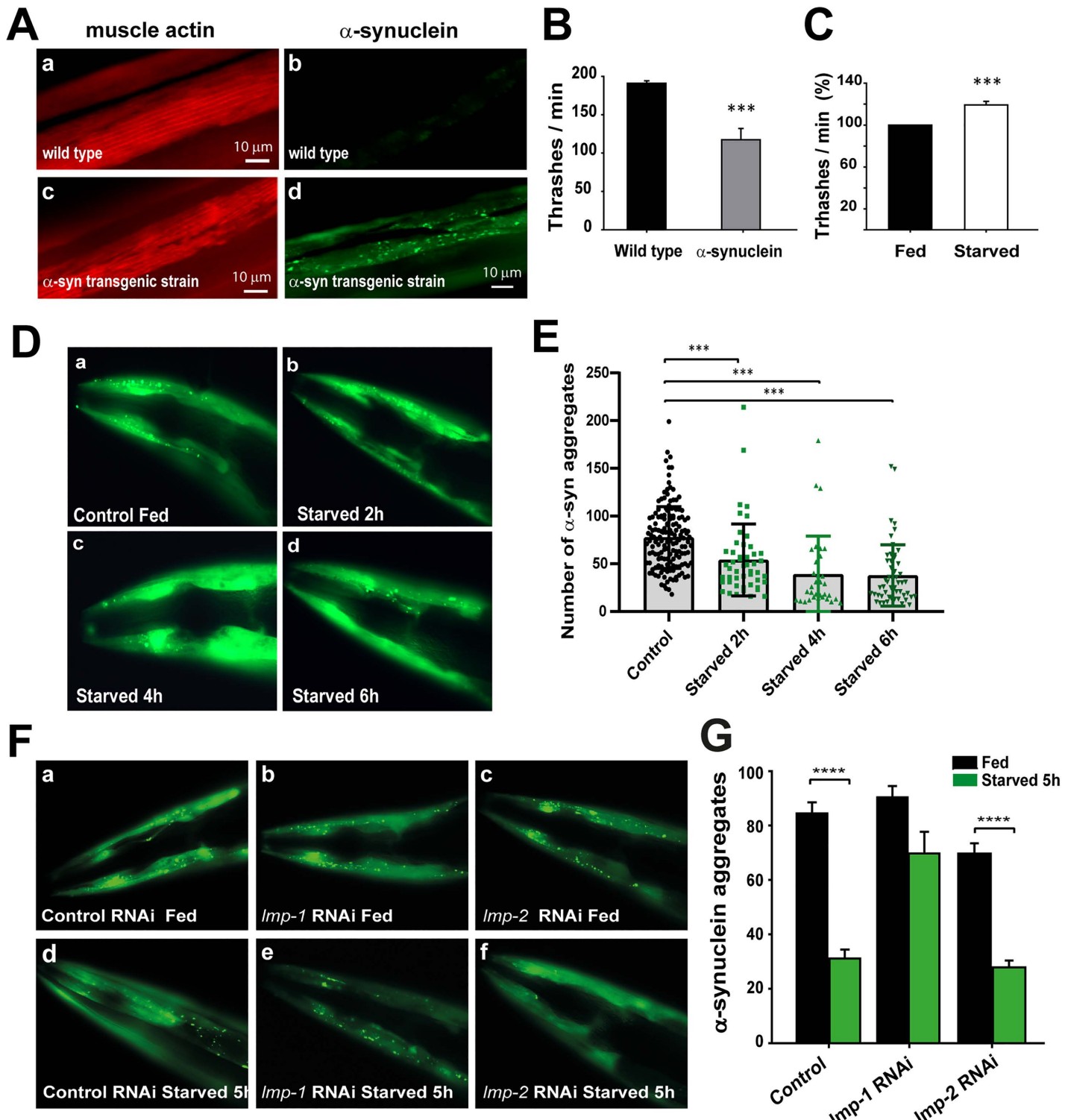

**Fig 4. Starvation improves locomotory capacity and decreases the number of α-synuclein-YFP aggregates of transgenic *C. elegans* through LMP-1 dependent mechanism.** A Fluorescence microscopy images of muscle cells from Wild type (a, b) and transgenic α-synuclein-YFP worms (NL5901) (c, d). Actin filaments for visualization of muscle fibers are stained with phalloidin-rhodamine (a, c). Florescence microscopy images show α-synuclein::YFP protein aggregates only in transgenic worms (b, d). Magnification 40X. B. Thrashing analysis of 2-day-old adult wild-type and

transgenic worms. C. Starvation for 4h improves locomotory capacity in the transgenic strain. D. Florescence images showing the heads of 2-day-old transgenic adult worms starved for 2, 4 and 6 h. E. α-synuclein aggregates in head muscle cells were quantified from fluorescence microscopy images using the ImageJ software. Graph represents mean±SEM of the number of aggregates present in the head muscle cells of at least 30 individuals per condition. F. Fluorescence images show the head regions of transgenic worms that express α-synuclein::YFP treated with control RNAi (a, d), RNAi against *lmp-1* (b, e) or *lmp-2* (c, f) under fed (a, b, c) or starved (d, e, f) conditions. G. Quantification of α-synuclein aggregate number per head from animals in figure F. The graph represents mean±SEM of the number of aggregates present in the head muscle cells of at least 30 individuals per condition ****p ≤ 0.0001 using Student's t-test.

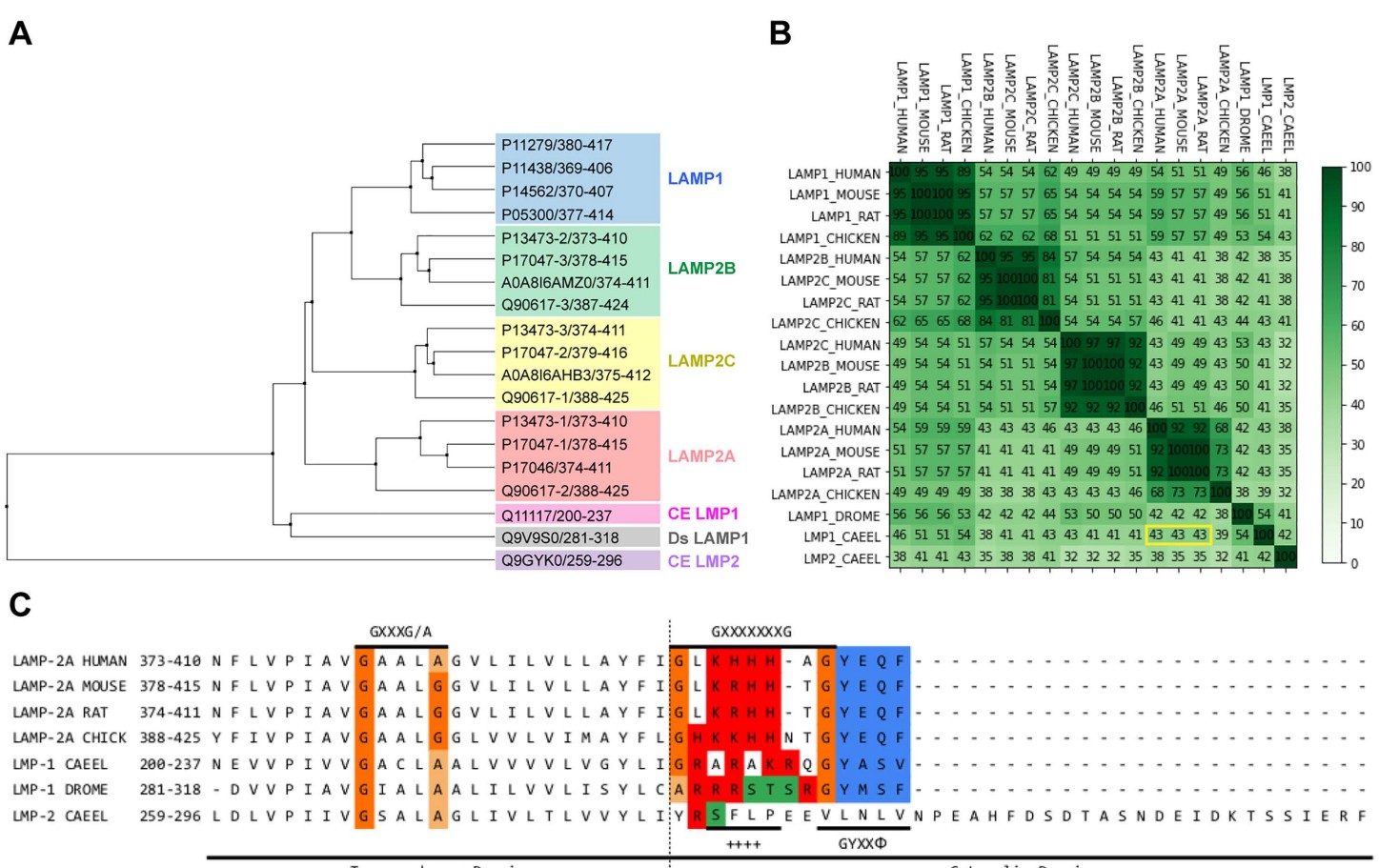

**Fig 5. Sequence analysis of transmembrane and C-terminal domains of *C. elegans* LMPs and comparison with LAMP orthologues.** A. Average distance tree of multiple sequence alignment between LAMP isoforms and LMP-1 and LMP-2. LAMP groups from transmembrane and C-terminal regions were labeled using human's sequence as reference. The figure shows *C. elegans* LMP-1 C-terminal sequences (CE LMP1) being closer to mammalian LAMP-2 isoforms when compared to *C. elegans* LMP-2 (CE-LMP2), being LAMP-2A the closest one. *D. Melanogaster* LAMP1 (Ds-LAMP1) is also included for comparison. B. Similarity matrix showing identity percentage between all sequences. Same isoforms show higher identity when compared to others, and LMP-1 has a higher identity to mammalian LAMP-2A than LMP-2 (values in yellow square). C. Detail of the alignment between *C. elegans* LMP1 (LMP-1_CAEEL) and LAMP2 (LMP-2_CAEEL), *D. Melanogaster* (LMP-1_DROME) and vertebrate LAMP2A orthologues. From left to right (N- to C-terminal) the motif for transmembrane homotypic interactions [GXXXG/A], the C-terminal cytosolic motif where important residues for LAMP2A/HSC70 are found [GXXXXXXXG], and the C-terminal end lysosome destination signal [GYXXΦ], are indicated. Positive residues in the [GXXXXXXXG] domain is shown in red.

at least 4 positively charged residues in the critical position determined to be necessary for interaction with HSC70 and KFERQ-containing substrates [47] (Fig 5C, in red, and Supporting information S5). The presence of Serine (S) and Threonine (T) residues at this position in the C-terminal tail, is a common feature of isoforms that have been shown not to be functional in CMA (Supporting information S5, in green). Interestingly, these residues are present in *C. elegans* LMP-2 and in the *D. melanogaster* LAMP-1 protein, but not in *C. elegans* LMP-1 (Fig 5C).

## Molecular modeling of the interaction of LMP-1 and LMP-2 with HSP-1

Our results indicate that both LMP-1 and HSP1 are necessary for association of KFERQ-containing proteins to lysosomes in *C. elegans.* Furthermore, analysis of the c-terminal sequence (GXXXXXXG cytoplasmatic domain) of LMP-1 identified the presence of positive residues that have been shown to be important for LAMP2A/HSC70 association and CMA activity in mammals [43]. To evaluate in silico if the C-terminal of LMP-1 can associate with HSP-1 similarly to what occurs in the LAMP2A/HSC70 interaction, we perform molecular modeling and docking studies. First, we constructed a 3D model of *C. elegans* HSP-1 based on available human and bovine crystal structures of HSC70 [22,23,24]. Interestingly, multiple sequence alignments between mammalian HSP70-like proteins and *C. elegans* HSP-1 indicate that this orthologue shows a higher homology with human HSC70 than to other human HSP70 proteins (Supporting information S4). We also modeled the structures of LMP-1 and LMP-2 transmembrane helices from the luminal hinge portion to the C-terminal based on the NMR structures of LAMP2A (Supporting information S5) [26]. For docking and molecular simulation of the hypothetical binding between *C. elegans* C-terminal LMP isoforms and HSP-1, we used representative structures from the more representative clusters of modeled proteins stabilized during the first 10 ns of the simulation (see Material and Methods). These clusters represent 93.8%, 82.7% and 72.2% of all structures obtained during the simulations of HSP-1, LMP-1 and LMP-2 structures, respectively. Docking of the LMP-1/HSP-1 shows LMP-1's C-terminal cytoplasmic residues inserted in a pocket on the nucleotide binding domain (NBD) of HSP-1 (Fig 5C and 6A, HSP1 NBD in green, LMP-1 in red) formed between the helices containing the residues ranging from 52 to 60 (Fig 6C, in cyan) and 258–270 (Fig 6C, in purple). For the LMP-2/HSP-1 complex, a similar binding site on HSP1 NBD as shown in LMP-1/HSP-1 complex was obtained, but with a change in the orientation of HSP-1 inducing a slight interference with the membrane (Fig 6B and 6D, HSP1 in green, LMP-2 in blue). On the other hand, the molecular docking simulation shows that, even though the interaction between HSP-1 with both proteins LMP-1 and LMP-2 remains stable during the full simulation (100 ns) (Fig 6 E and 6F, and Supplementary Video 1), the complex HSP-1/LMP-2 (Fig 6E, in blue) shows higher structural fluctuations than the HSP-1/LMP-1 complex (Fig 6E, in red). Moreover, higher variations in the radius of gyrus were observed between HSP-1/LMP-2 complex (Fig 6F, blue line), suggesting that the complex of HSP-1 with LMP-1 may be more stable (Fig 6F, red line).

NMR based structural studies of the secondary and tertiary structure of LAMP2A transmembrane domains reconstituted in micelles, indicate the existence of a stable oligomeric structure composed of three molecules wrapped around each other to form a parallel coiled coil trimer [26]. This oligomeric form of LAMP2A, observed also in membranes of isolated liver lysosomes, and the presence of transmembrane GXXXG/A motifs have been shown to be important for homophilic interactions between transmembrane helices [48] and for the formation of higher order LAMP2A oligomers able to translocate CMA substrates into the lysosome for degradation [49]. Transmembrane GXXXG/A motifs are conserved in LMP-1 and LMP-2 *C. elegans* proteins, however, in order to evaluate if LMP-1 or LMP-2 are able to form oligomers *in silico* we modeled and compared the structure of full-length proteins based on a template model of LAMP2A trimeric structure in a virtual lipid bilayer based on the lipid composition of lysosomal membranes (Fig 7) [26,50]. The results indicate that intrinsically, both trimeric structures are stable during the dynamics (100 ns) (Supplementary Video 2), suggesting that, at least at the structural level, *C. elegans* LMP-1 and LMP-2, could associate similarly as mammalian LAMP2A.

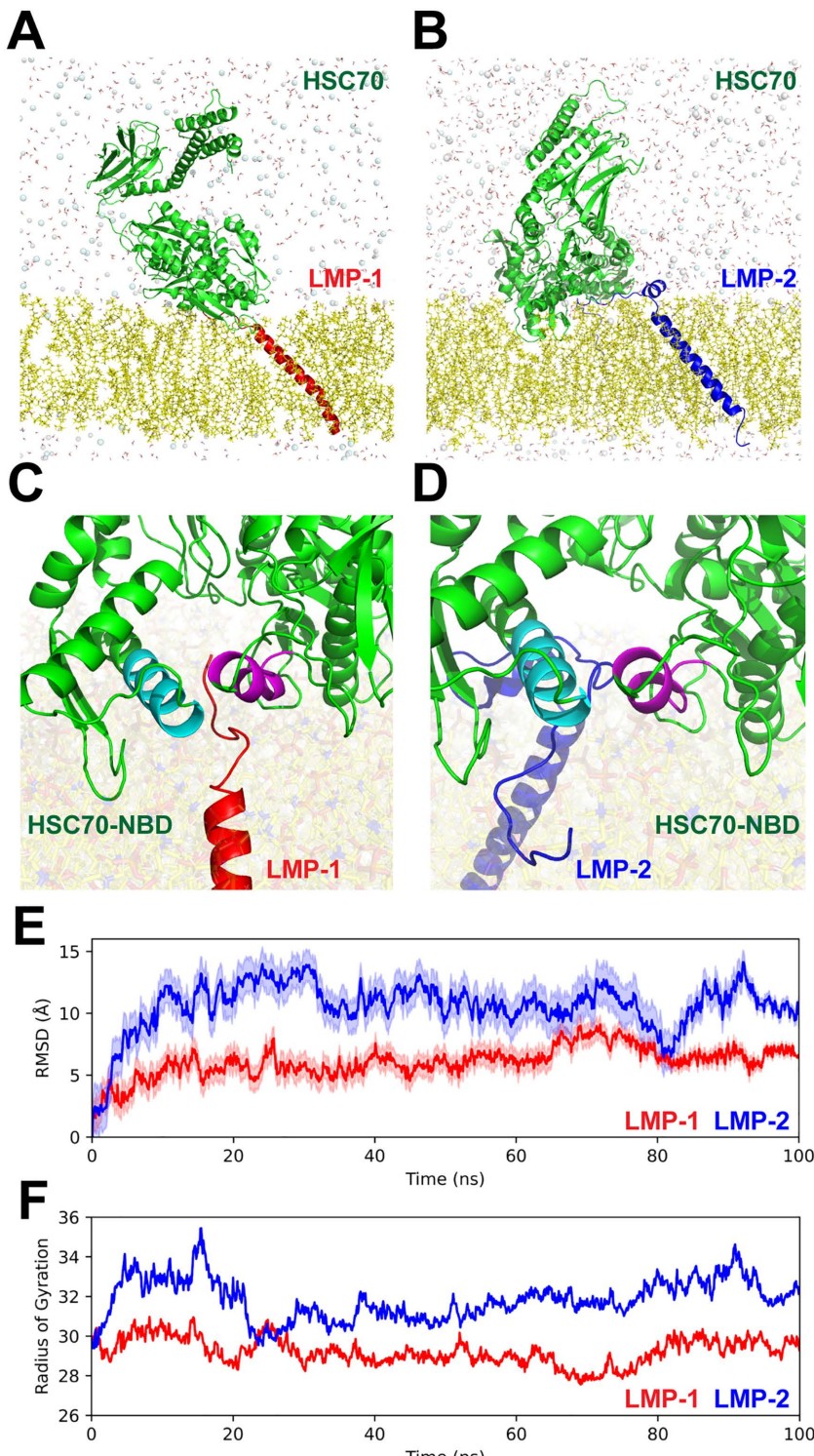

**Fig 6. Molecular dynamics of the LMP-1/HSP-1 and LMP-2/HSP-1 complex models.** A-D Structure of complex models of (A, C) LMP-1 (in red) and (B, D) LMP-2 (in blue) in complex with HSP-1 (green) inserted in a POPC membrane (yellow) generated through protein-protein docking using the ClusPro server. C-D Close-up of the predicted interaction site of the C-terminal tail of LMP-1 (in red) and LMP-2 (in blue) with NBD domain of HSC70 (in green). Identified interacting helices in the HSP1/LMP-1 complex ranging from 52 to 60 and 258 to 270 are colored in cyan and purple. E) Graph of RMSD fluctuations (lines) and their respective standard deviations (shaded line) and F) calculated radius of gyration derived from molecular dynamic simulation box for LMP-1/HSP-1 complex (in red) and LMP-2/HSP-1 (in blue).

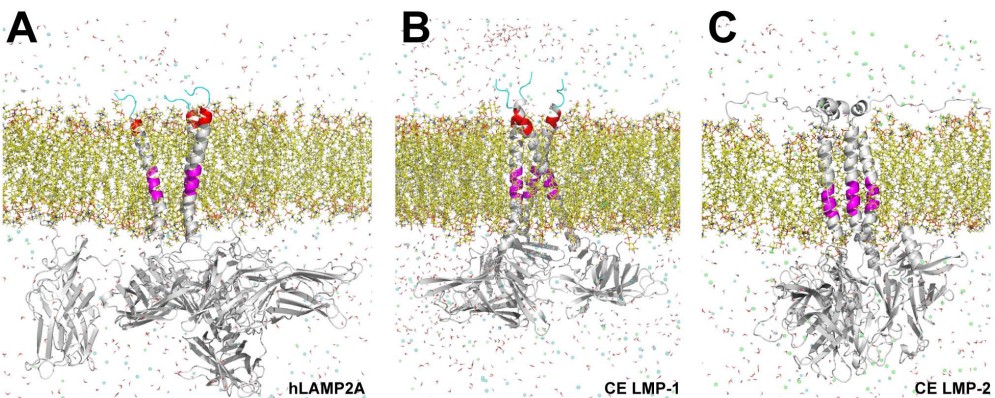

**Fig 7. Molecular dynamics of LAMP2A, LMP-1 and LMP-2 trimers.** Structure of trimer models of LAMP2A, LMP-1 and LMP-1 (A-C) inserted in a lysosomal membrane (yellow). Highlighted in magenta are the motifs for transmembrane homotypic interactions [GXXXG/A], in red are the C-terminal positively charged residues in the cytosolic motifs [GXXXXXXXG], and in cyan the C-terminal end lysosome destination signal [GYXXΦ].

## Discussion

Using a fluorescent protein reporter with KFERQ sequences, we demonstrated starvation-induced specific association of KFERQ proteins with lysosomes in *C. elegans* muscle, resembling mammalian chaperone-mediated autophagy (CMA) [32,39]. Fluorogenic Cathepsin D substrates showed KFERQ proteins colocalizing with acidic lysosomes, with clear colocalization in regions of high enzyme activity, unlike diffuse patterns seen in some mammalian cells [51]. A control mCherry lacking KFERQ sequences did not colocalize with lysosomes, confirming the specificity of the KFERQ motif. Interestingly, while most PAmCherry puncta colocalized with lysosomes, not all cathepsin-positive lysosomes showed KFERQ-protein association, similar to findings in mammals where only a fraction of LAMP2A and HSC70 positive lysosomes show increased KFERQ-protein degradation [32,43,52].

Selective KFERQ-dependent recruitment of cytosolic proteins to lysosomes occurs via chaperone-mediated autophagy (CMA) and microautophagy (MiA). It has been suggested that a LAMP2A-dependent CMA process is absent in invertebrates due to the lack of a 2A exon in their LAMP genes [11,41]. Comparisons of LAMP protein sequences across species further indicate the absence of a functional LAMP2A orthologue, implying that KFERQ-selective CMA-like autophagy may not exist in invertebrates [10,11,40,41]. However, a LAMP-independent KFERQ-selective microautophagy, dependent on ESCRT machinery, has been identified in *Drosophila* [13,40]. The presence of selective KFERQ autophagy in *C. elegans* remains uncharacterized.

We observed strong KFERQ-dependent association of a fluorescent reporter to lysosomes in *C. elegans* muscle during long-term starvation. Interestingly, while mammalian chaperone-mediated autophagy (CMA) is highly activated under nutrient deprivation, endosomal microautophagy (eMI) remains constitutive and is inhibited by starvation [53]. Mammalian eMI occurs in late endosomes, selectively degrading KFERQ-containing cytosolic proteins via Hsc70 and ESCRT, but does not involve secondary lysosomes [53,54]. In *Drosophila*, MiA is active in lysosomes, induced by starvation and mTOR inhibitors, and independent of LAMP, as KFERQ localization persists in *Drosophila Lamp1* mutants [13,40,55].

In this work, we identified a KFERQ-selective pathway in lysosomes of *C. elegans* that is highly responsive to nutrient deprivation. Consistent with other reports [46,56], deletion of LMP-1 does not impact lysosome integrity or activity, but it does affect the KFERQ-dependent autophagy response to nutrient deprivation, unlike LMP-2. This result indicates a specific role for LMP-1 in the KFERQ-selective autophagy pathway.

Inhibition of macroautophagy stimulates chaperone-mediated autophagy (CMA), while CMA activation inhibits macro-autophagy [57–59]. This crosstalk has not been observed between microautophagy (MiA) and macroautophagy, although MiA may degrade macroautophagy receptors, potentially regulating the switch between selective and bulk autophagy [1]. In *C. elegans,* LGG-1 and LGG-2 are non-redundant ATG8 homologs involved in macroautophagy, with LGG-1 being essential for aggrephagy and LGG-2 supporting autophagosome maturation and selective macroautophagy [60]. Our findings indicate that knockdown of LGG-1 and LGG-2 does not alter KFERQ-PAmCherry levels under fed conditions; however, LGG-1 knockdown enhances the KFERQ-dependent autophagy response to starvation. Autophagosomes in *C. elegans* appear after 20 minutes of starvation and can be detected for up to 4 hours [36,37,61]. CMA is significantly acti-vated after only 4 hours of nutrient deprivation; however, under macroautophagy inhibition, CMA sensitivity increases at 2 hours, suggesting an inverse crosstalk, like that observed between CMA and macroautophagy in mammals [57,58].

To evaluate the role of LMP-1 in chaperone-mediated autophagy (CMA), we investigated its effect on human α-synuclein levels in *C. elegans* muscle. Soluble wild-type (WT) α-synuclein is primarily degraded by CMA, and impaired CMA is linked to Parkinson's disease (PD) pathogenesis [62]. The evidence suggests that CMA plays a crucial role in maintaining basal levels of soluble α-synuclein and in aggregate formation when degradation mechanisms are inhibited, often surpassing macroautophagy (MA) in this function [63]. In mammalian cells, enhancing CMA through LAMP2A over-expression or retinoic acid derivatives reduces total α-synuclein levels and aggregates [64]. Additionally, HSC70 has been shown to interact preferentially with high molecular weight α-synuclein assemblies, altering their properties and reducing toxicity by removing monomeric units from fibril ends [65,66]. Our results demonstrate a strong dependency on LMP-1 for the starvation-induced reduction of α-synuclein aggregates in *C. elegans* muscle.

Macroautophagy plays a critical role in degrading protein aggregates in *C. elegans*, especially in the selective degrada-tion of P granules and SQST-1-positive aggregates (p62 in mammals) [67]. LGG-1 and LGG-2, homologs of the mamma-lian ATG8, have distinct functions in autophagosome formation and cargo recognition. LGG-1 is essential for aggregate degradation, while LGG-2 has cargo- and developmental-stage-specific roles [60]. However, the role of *lgg-1* and *lgg-2* in regulating exogenous aggregates like α-synuclein remains to be fully validated in *C. elegans* [68,69].

The degradation of α-synuclein involves both the ubiquitin-proteasome system (UPS) [70] and autophagy-lysosomal systems (ALS), including macroautophagy [38], chaperone-mediated autophagy (CMA) [71], and HSC70-independent endo-somal microautophagy [72]. The route of α-synuclein degradation depends on factors like cell type, α-synuclein variant (wild-type or mutant), and its ubiquitination/aggregation status [73–76]. *In vivo* mammalian data suggest lysosomes as the primary pathway for α-synuclein clearance [77,78]. Loss of LAMP2A correlates with increased α-synuclein levels in early-stage PD, and CMA deficiencies contribute to α-synuclein aggregation and neurodegeneration in α-synuclein-overexpressing models, supporting CMA's role in α-synuclein regulation and PD pathogenesis [63,79,80]. CMA degrades wild-type α-synuclein mono-mers/dimers but not mutant or oligomeric forms [38,77,81]. HSC70 can disassemble α-synuclein amyloids, and its knock-down increases monomeric and oligomeric α-synuclein in human neuronal cells [82–84].

Protection against proteotoxicity induced by aggregate-prone proteins in *C. elegans* muscle depends on lysosomal gene expression, including *lmp-1,* but is independent of macroautophagy genes [15]. Similarly, protein aggregate clear-ance in nematode germ cells remains unaffected by macroautophagy gene knockdown, indicating lysosomal involvement in directly engulfing protein aggregates or their precursors [16]. Studies on the degradation of RHO-1 and KIN-19 in *C. elegans* pharyngeal muscle cells identified a lysosome-dependent, macroautophagy-independent pathway, SAPA (Safe-guard Against Protein Aggregation), which degrades soluble aggregation-prone proteins before aggregates form [17,85]. While *lmp-1* is not required for macroautophagy in *C. elegans*, LAMP-2 deficiency in mammals causes severe autophago-some accumulation [86], and LAMP-1 is dispensable for macroautophagy in *Drosophila* [40]. ESCRT-dependent KFERQ microautophagy in *Drosophila* is also independent of Lamp1 [13,40]. However, our results suggest *lmp-1* works with HSP-1 chaperone in KFERQ-selective autophagy in *C. elegans.* Moreover, *lmp-1* deficiency blocks the effects of HSP-1 knockdown, implying a shared mechanism.

In mammalian chaperone-mediated autophagy (CMA), HSC70 interacts with specific residues in the C-terminal tail of LAMP2A, facilitating substrate association, LAMP2A oligomerization, and substrate translocation through LAMP2A pores into the lysosome lumen [1]. Our findings suggest that HSP-1 and LMP-1 may operate in a similar molecular interaction for KFERQ-selective autophagy in *C. elegans,* prompting us to evaluate whether LMP-1 serves as a docking site at the lysosomal membrane, akin to LAMP2A's role in mammals. Although LMP-1 is considered an orthologue of human LAMP1/2 based on BLAST alignment, and LMP-2 has been associated with mammalian LAMP2A functionality, comparative analysis of full-length amino acid sequences indicates a lack of a functional LAMP2A orthologue in invertebrates [10,11,14,40–42]. Notably, our results reveal that LMP-1 exhibits higher identity with mammalian LAMP2A than LMP-2 in the C-terminal and transmembrane domains, critical for HSC70 recognition and pore formation [47]. Although *C. elegans* LMP-2 contains additional amino acids at its C-terminus, this does not negate its classification as an orthologue of LAMP2A since extra residues do not hinder lysosomal targeting or KFERQ substrate recognition [11]. Importantly, positively charged residues necessary for HSC70 and KFERQ substrate interaction are present in LMP-1 but absent in LMP-2 [47]. The positively charged motifs in mammalian LAMP2A orthologues include lysine (K), arginine (R), and histidine (H), which are crucial for HSC70 binding [11,87]. Interestingly, the LMP-1 C-terminal sequence (GRARAKRQG) contains four additional positively charged residues, compared to LMP-2 (Fig 5). Bioinformatics analyses suggest that HSP-1 interacts more stably with the C-terminus of LMP-1 than with that of LMP-2, emphasizing LMP-1's potential role as a docking site for HSP-1 in mediating KFERQ-selective autophagy in *C. elegans.* Further experimental studies are necessary to confirm the direct interaction between these proteins and elucidate their specific roles in this pathway.

A crucial property of CMA related to LAMP2A is the formation of a transient translocation pore, resulting from the dynamic association of LAMP2A in oligomeric complexes [88]. This oligomerization is facilitated by homomeric associations of transmembrane GXXXXG/A motifs and specific interactions between luminal domains through β-prism fold structures [87,89,90]. Both LMP-1 and LMP-2 possess these transmembrane motifs, which are essential for homophilic interactions and the formation of higher-order LAMP2A oligomers [26,48,49]. Our findings indicate that trimeric models of both LMP-1 and LMP-2 are stable. We used these trimeric models for molecular dynamics studies based on the observation that mammalian LAMP2A forms stable coiled-coil trimers capable of binding HSC70 and CMA substrates [26].

Taken together, our results suggest the existence of a of KFERQ-dependent selective autophagy mechanism in *C. elegans*. Our *in vivo* results and bioinformatic analyses indicate that LMP-1 and HSP-1 have an active role in this pathway equivalent to the role of LAMP2A and HSC70 in mammals. This finding opens opportunities to study, in this simpler model organism, complex mechanisms of protein degradation control under both physiological and pathological conditions.

## Supporting information

**S1 Fig. Masks of quantified particles generated from representative microscopic images shown in** Fig 1C **using ImageJ software.** (A) Representative image of fed worms (corresponding to Fig 1C, panel a). (B) Representative image of starved worms (corresponding to Fig 1C, panels b–d). Masks were created by thresholding and particle analysis in ImageJ to quantify puncta associated with the KFERQ reporter. All images were processed under the same parameters for accurate comparison.
(TIF)

**S2 Fig. A. Control of autofluorescence background and crosstalk of SiR700-lysosome probe fluorescence in the KFERQ-PA-mCherry channel (Channel2: excitation: 543 nm; emission: 572 ± 20 nm) in non-photoactivated worms.** a) Confocal image of a worm head stained with the Sir700-lysosome probe (CH2; excitation: 635 nm; emission: 655–755). b) Confocal image of a worm head from a non-photoactivated animal stained with SiR700-lysosome probe observed in the mCherry channel (CH1: excitation: 543 nm; emission: 572 ± 20 nm). c) The image in b with enhanced contrast using histogram equalization to show signal presence in b. Scale bar = 20 µm. B. Comparison of the signal intensity

in Sir700-lysosome fluorescence channel (lysosomal probe) vs. KFERQ-PAmCherry fluorescence. a) Image of a worm's head stained with SiR700-lysosome (CH2; excitation: 635 nm; emission: 655–755) and b) corresponding pseudocolored image. c) Image of a photoactivated worm expressing KFERQ-PA-mCherry observed in the Sir700 channel (CH2; excitation: 635 nm; emission: 655–755), and d) corresponding pseudocolored image acquired using the same settings as in a and b. C. Intensity correlation analysis of KFERQ-PAmCherry with the lysosomal activity marker SiR700-lysosome. a) Representative image of muscle tissue from the head region showing the KFERQ-PAmCherry reporter (red) following photoactivation and incubation with SiR700- lysosome (green). Yellow puncta show areas of colocalization between the signals from both channels, indicating overlap between KFERQ-PAmCherry and lysosomes. b) Pseudocolored image showing the correlation of signal intensities between KFERQ-PAmCherry and SiR700-Lysosome. Plots c and d show the degree of synchronization of the KFERQ-PAmCherry and SiR700-Lysosome signals (products of the differences of the intensity of each pixel with respect to the mean intensity of each channel) normalized with respect to the SiR700-Lysosome and PAmCherry signal.
(TIF)

**S3 Fig. PAmCherry aggregates in the absence of KFERQ do not colocalize with the lysosomal marker.** A) Representative confocal images of 2-day old transgenic adult worm of expressing PAmCherry in muscle cells. mCherry (red 572/50 emission) and the Lys700 lysosomal SiR700-lysosome (green). (a) PAmCherry reporter fluorescence in head muscles (b) SiR700-lysosome fluorescence image (green). (c) Merge of the images in a and b, yellow indicates colocalization of the mCherry reporter with lysosomes. (d) Zoom of the region labeled in (a), (b) and (c). B) Intensity correlation analysis of PAmCherry with lysosomal activity. (a) Representative image of nematode head muscle tissue in which the PAmCherry reporter (red) was photoactivated and incubated with SiR700-lysosome (green). (b) Pseudocolored image showing the correlation of signal intensities between PAmCherry and SiR700-lysosome. Plots (c) and (d) show the degree of synchronization of the PAmCherry and SiR700-lysosome signals (products of the differences of the intensity of each pixel with respect to the mean intensity of each channel) normalized with respect to the SiR700-lysosome (c) and PAmCherry (d) signal. (delta KFERQ).
(TIF)

**S4 Fig. Average distance tree of HSP70-like proteins.** Average distance tree built with a multiple sequence alignment preformed with ClustalW.
(TIF)

**S5 Fig. Multiple sequence alignment of LAMP2 isoforms and LAMP1.** All LAMP2 isoforms and LAMP1 are following the same order (human, mouse, rat and chicken). Name on the right is based on the nomenclature used for human sequences. From left to right (N- to C-terminal) the motif for transmembrane homotypic interactions [GXXXG/A], the C-terminal cytosolic motif where important residues for LAMP2A/HSC70 are found [GXXXXXXXG], and the C-terminal end lysosome destination signal [GYXXΦ], are indicated. Positive residues in the [GXXXXXXXG] domain is shown in red.
(TIF)

**S1 Video. 100 ns molecular dynamics of HSP-1/LMP-1 and HSP-1/LMP-2 complexes.** HSP-1 is depicted in green, LMP-1 in red, LMP-2 in blue and lysosomal membrane in yellow. For easier visualization water molecules were removed and Na$^+$ and Cl$^-$ atoms represented as white dots.
(MP4)

**S2 Video. 100 ns molecular dynamics of trimers.** Structures of human LAMP2A, and C. elegans LMP-1 and LMP-2 are depicted in silver, lysosomal membrane in yellow. For easier visualization water molecules were removed and Na$^+$ and Cl$^-$ atoms represented as white dots.
(MP4)

## Author contributions

**Conceptualization:** Alicia N Minniti, Rebeca Aldunate, Ivan E Alfaro.

**Data curation:** Maria Gallardo-Campos, Alicia N Minniti, Rebeca Aldunate, Ivan E Alfaro.

**Formal analysis:** Maria Gallardo-Campos, Alicia N Minniti, Juan Hormazabal, Gonzalo Nuñez, Carlos F Lagos, Rebeca Aldunate, Ivan E Alfaro.

**Funding acquisition:** Ivan E Alfaro.

**Investigation:** Maria Gallardo-Campos, Gonzalo Nuñez, Carlos F Lagos, Rebeca Aldunate, Ivan E Alfaro.

**Methodology:** Maria Gallardo-Campos, Alicia N Minniti, Juan Hormazabal, Gonzalo Nuñez, Carlos F Lagos, Tomas Perez-Acle, Rebeca Aldunate, Ivan E Alfaro.

**Supervision:** Juan Hormazabal, Rebeca Aldunate.

**Validation:** Ivan E Alfaro.

**Writing – original draft:** Alicia N Minniti, Rebeca Aldunate, Ivan E Alfaro.

**Writing – review & editing:** Rebeca Aldunate.

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
