## [Decision Letter · Decision Letter 0]

13 Jan 2025

PONE-D-24-53431KFERQ-selective protein autophagy in Caenorhabditis elegans depends on LMP-1PLOS ONE

Dear Dr. Aldunate,

Thank you for submitting your manuscript to PLOS ONE. After careful consideration, we feel that it has merit but does not fully meet PLOS ONE’s publication criteria as it currently stands. Therefore, we invite you to submit a revised version of the manuscript that addresses the points raised during the review process.

We look forward to receiving your revised manuscript.

Kind regards,

David Chau

Academic Editor

PLOS ONE

Journal Requirements:

“National Agency for Research and Development (ANID); ANID [Centro Ciencia & Vida, FB210008, Financiamiento Basal para Centros Científicos y Tecnológicos de Excelencia to IEA  and CFL]; ANID [FONDECYT 11161056 to IEA]; ANID [ FONDECYT Posdoctorado 2021- 3210596 to GN]. Powered@NLHPC: This research was partially supported by the supercomputing  infrastructure of the NLHPC (CCSS210001). Some strains were provided by the CGC, which is funded by NIH Office of Research Infrastructure Programs (P40 OD010440)”

“National Agency for Research and Development (ANID); ANID [Centro Ciencia & Vida, FB210008, Financiamiento Basal para Centros Científicos y Tecnológicos de Excelencia to IEA  and CFL]; ANID [FONDECYT 11161056 to IEA]; ANID [ FONDECYT Posdoctorado 2021- 3210596 to GN]. Powered@NLHPC: This research was partially supported by the supercomputing  infrastructure of the NLHPC (CCSS210001). Some strains were provided by the CGC, which is funded by NIH Office of Research Infrastructure Programs (P40 OD010440)”

6. We notice that your supplementary figures are uploaded with the file type 'Figure'. Please amend the file type to 'Supporting Information'. Please ensure that each Supporting Information file has a legend listed in the manuscript after the references list.

Reviewers' comments:

Reviewer's Responses to Questions

**Comments to the Author**

1. Is the manuscript technically sound, and do the data support the conclusions?

Reviewer #1: Partly

Reviewer #2: No

2. Has the statistical analysis been performed appropriately and rigorously? 

Reviewer #1: I Don't Know

Reviewer #2: I Don't Know

3. Have the authors made all data underlying the findings in their manuscript fully available?

Reviewer #1: Yes

Reviewer #2: No

4. Is the manuscript presented in an intelligible fashion and written in standard English?

Reviewer #1: Yes

Reviewer #2: Yes

5. Review Comments to the Author

Reviewer #1: In this study, authors present original results on C. elegans chaperone-mediated autophagy. They generated an in vivo model to monitor chaperone-mediated autophagy (CMA) in body wall muscles, based on a previously described KFERQ-bearing photoactivable mCherry reporter. In the first part, they show that starvation triggers KFERQ-PAmCherry localization to lysosomes and that LMP-1 but not LMP-2 is required for such localization to lysosomes upon starvation. In addition, they report that LMP-1 but not LMP-2 is required for selective autophagy of human α-synuclein in body wall muscles of C. elegans. In the second part (which is not of my expertise), using bioinformatics and molecular modeling they suggest that LMP-1 but not LMP-2, is the potential orthologue of LAMP2A in C. elegans and that an interaction between LMP-1 and HSP-1 is predicted to be stable.

CMA has established physiological relevance in protein quality control and its failure is associated with age-associated diseases (Kaushik and Cuervo, Nat Rev Mol Cell Biol, 2018). Indeed, CMA research in C. elegans is very limited, with only a few publications suggested superficially that CMA is functional in C. elegans to my knowledge (Regitz et al, Eur J Nutr, 2016 and Eisermann D.J. et al, Biochem Biophys Res Commun, 2017). For these reasons, I find the fields of selective types of autophagy and CMA of exceptional importance and of broad interest in the community of cell biology and I think that the aim of this study falls into the above prospects.

The manuscript needs careful revision concerning the terminology in the field of autophagy and others. There are several neglected points in terms of grammar, accuracy and consistency. Some of the authors’ conclusions are not entirely supported by the data presented. Importantly, the quality of some images is not of high quality. Specific examples for the above, but not an exhausted list, are provided in the comments.

Major comments

1. The quality of several fluorescent images is poor. Examples include Fig. 1C, 2A, 2C, 2E, 2F, 3A and 4D. Higher resolution is needed to clearly demonstrate the punctate structures.

2. Several fluorescent images are possibly overexposed. Examples include 1C, 2A, 2F, 3A and 4D.

3. The above two points make the demonstration of the puncta in the images and the correlation to the respective quantifications difficult. For example, which particles were counted as puncta in Fig. 1Cd showing approximately 25 KFERQ-PAmCherry puncta in 6h starvation? Which particles were counted as puncta in Fig. 4D showing approximately 40 to 80 α-synuclein aggregates in these conditions?

4. Fig. 1E: It seems there is plenty of some kind of fluorescent dirt, possibly dye remnants, obvious especially in the mCherry channel. What are those spots? Importantly, two arrowheads point to out of focus cells. The arrowhead inside in one of the cells in focus points to no puncta structure. The above make the conclusion “only a fraction of these lysosomes can associate the KFERQ fluorescent protein” questionable (lines 264-265).

5. Sup. Fig. 2A: It seems that the majority of the lysosomal marker is not absorbed at the PA-mCherry-expressing muscles cells of the head. The two signals somehow anticorrelate in whole cell level. How the colocalization studies took place? It would be informative if each single and merged fluorescent image were provided in all figures apart from the possible zoomed areas.

6. Lines 302-304: The authors claim that during the 2h starvation period an LC3-independent selective autophagy might be active in the case of lgg-1 knockdown. Why this is specific to lgg-1 knockdown? Why there is a no increase of KFERQ-PAmCherry localization to lysosomes during the 2h starvation in the control(RANi) or lgg-2(RNAi)?

7. Fig. 4C: Wild type animal controls of the same conditions are missing. Do wild type animals show an increase in thrashes upon 4h starvation as well?

8. Fig. 4: The thrashing assay (Fig. 4C) predominantly reflects defects in the body wall musculature of C. elegans. The differences in α-synuclein aggregates in head muscles (Fig. 4D, E) cannot be easily correlated with the thrashing assay (lines 322-323).

9. Even if in the “Data and statistical analysis” section it is mentioned that “The specific statistical tests used in each experiment are indicated in the corresponding Figure Legends” I could not find any of them.

Minor comments

10. To my knowledge SAR is a well-established term that stands for “selective autophagy receptors” and not “specialized autophagy receptors” (line 69). Authors’ citation 5 uses the term selective autophagy receptors as well.

11. The result title “Starvation triggers KFERQ-PAmCherry destination to lysosomes in C. elegans muscle cells” needs correction.

12. Fig. 1E legend: (a) is not described and (d) is the merge of the boxed regions from (b) and (c). Whole plane merge image is not provided.

13. The phrase “Composite merging…” contains redundant information (Sup. Fig. 2Ac legend).

14. The word “dots” could be replaced by the word “puncta” in the text and the figures for consistency and academic language usage.

15. Line 140: Replace “nr°2045” with “nr2045”.

16. Line 175: Correct syntax.

17. In C. elegans the L1 (Larval 1) stage is a post-embryonic developmental stage and not an “embryo L1 stage” (lines 186-187). Replace “feed” with “fed”.

18. Lines 189-192: Why did the photoactivation in the 96-well optical plates take place for 5-10min and not for a strictly fixed period? Could this affect the amount of photoactivated protein along with the experimental results?

19. The term “transfected” is not appropriate for genetically modified (transgenic) C. elegans (line 210).

20. Fig. 1D legend: Typically, three asterisks (***) represent a p value ≤0.001 and not ≤0.0001. Please correct accordingly.

21. Sup. Fig. 1B and Sup. Fig. 2Bb: The heatmaps with the correlation values are missing.

22. Fig. 2B: There are different sized points at the last three columns. Is there statistical significant difference between starved control(RNAi) and hsp-1(RNAi) as it is implied in lines 273-275 and 292-293. Please include such statistical analysis.

23. Keep consistent axis labelling. For example, “Puncta number per head” (Fig. 2B, G) and “Puncta number / head” (Fig. 2D).

24. Fig. 2C: In my eyes (a) and (b) looks like different planes of the same animal.

25. Lines 292-297: Since there is no increase in the number of KFERQ-PAmCherry puncta under starvation in control lmp-1 mutant animals (Fig. 2F, G), the claim “in the lmp-1 knock out mutant background this effect (decrease in the number of KFERQ-PAmCherry puncta under starvation upon knock-down of hsp-1) was not observed” can not be supported. Actually, it seems there is a tendency of an increase in the number of KFERQ-PAmCherry puncta. Statistics are needed. It makes sense to show that some comparisons show non-significant statistical difference.

26. It seems there is high discrepancy of the KFERQ-PAmCherry puncta numbers in identical conditions between experiments. For example at the fed control(RNAi) condition between Fig. 2B (close to 15) and Fig. 3B (close to 4).

27. Line 302: To my knowledge lgg-1 is an ortholog of and mostly related to GABARAP while lgg-2 to LC3.

28. Fig. 4G: Statistics between RNAi conditions are missing.

29. Lines 441-443: The results of Fig. 2F and G show that knockdown of hsp-1 in the lmp-1 background does not exacerbate the phenotype under starvation. On the contrary, it seems that hsp-1 knockdown tends to rescue the lmp-1 mutant phenotype by restoring the KFERQ-PAmCherry puncta formation upon 4h starvation.

30. Does reference 59 appears somewhere in the text?

31. What does magnification 400x mean? Typically, description of the objective magnification is enough.

Reviewer #2: In this manuscript, the authors are trying to establish that LMP-1 is the homolog/ equivalent of human LAMP-2A to bring about chaperone mediated autophagy through the lysosomal targeting KFERQ tagged with the photoactivatable fluorescent reporter, mCherry expressed in the(head) muscle cells of C. elegans. They have carried out feeding based RNAi to provide evidence for the increase/decrease of lysosomal punta for C. elegans lmp-1, lmp-2 and lmp-1 mutant, and the chaperone, heat shock protein, hsp-1. Further, they have done RNAi to rule out involvement of macroautophagy with LGG-1 and LGG-2. In addition, they show that the Parkinson causing protein, alpha- synuclein expression in the muscle cells, leading to reduction in thrashing as aggregates accumulate and it is degraded through chaperone mediated autophagy. Further, they have tried modeling and homology studies to show that LMP-1 is closer to mammalian LAMP2A. But, though lot of work has been done, the provided data is not convincing enough for the claims.

The major claim is based on the changes in (lysosomal) puncta upon starvation in the muscle cells of the head. Except in the lmp-1mutant (Fig. 2E) and the supplementary Fig. S2, the puncta is not visible in the images. In addition, in lmp-1 mutant, there is no significant difference between the wildtype and lmp-1 mutant. If so, how did they quantity the puncta ? Appropriate punta containing images need to be provided to validate their claims.

6. PLOS authors have the option to publish the peer review history of their article (what does this mean? ). If published, this will include your full peer review and any attached files.

**Do you want your identity to be public for this peer review?** For information about this choice, including consent withdrawal, please see our Privacy Policy .

Reviewer #1: No

Reviewer #2: No

---

## [Author Response · Author response to Decision Letter 1]

13 Mar 2025

1. Files were uploaded with the required file names.

2. We fixed the information and provided grant number in the Funding Information section.

3. We added the required statement below.

4. Find below the amended Funding Statement as detailed in the guide for authors, and added the required statement as follows:

National Agency for Research and Development (ANID), https://anid.cl/: Financiamiento Basal para Centros Científicos y Tecnológicos de Excelencia #FB210008 to IEA and CFL, FONDECYT#11161056 to IEA, and FONDECYT Posdoctorado 2021- #3210596 to GN. This research was partially supported by the supercomputing infrastructure of the NLHPC (#CCSS210001), https://www.nlhpc.cl/. Some strains were provided by the CGC https://cgc.umn.edu/, which is funded by NIH Office of Research Infrastructure Programs (P40 OD010440).

There was no additional external funding received for this study. The funders had no role in study design, data collection and analysis, decision to publish, or preparation of the manuscript.

5. We included a separate caption for each figure in the manuscript.

6. The supplementary figures´ file types were changed to “Supporting Information”

7. We did not use LaTeX to create our manuscript.

Answer to the reviewers’ comments:

2. The statistical test used and the sample size for each experiment are indicated in the figure legends.

3. Data was uploaded to https://dataverse.harvard.edu/dataset.xhtml?persistentId=doi:10.7910/DVN/DFPZQX.

Review comments to the author

Reviewer #1

Major comments:

1 y 2. The figures were originally submitted in pdf format and were therefore of very low quality. We have now submitted the original .eps files.

With respect to the possible overexposure of some photographs: Figure 1C: all photographs were taken with the same magnification and exposure, which allowed us to detect the distribution pattern of the fluorescent marker using an epifluorescence microscope (not confocal). With this exposure we can show that the florescent marker is localized in the cytosol under fed conditions, while under starvation conditions the pattern shifts to a more punctuated pattern over time. In the new version, this pattern can be seen more clearly, especially if the figure is zoomed in. This explanation is also valid for figures 2A, 2F and 3A.

3. All relevant information about the protocol for KFERQ-PAmCherry punta number measurement was added to the Material and Methods section as follows (from line 214-221): briefly, Image analyses of the puncta number in the worms’ head region were performed using the ImageJ software. The images were converted into 8-bit format, contrast was enhanced, and KFERQ-PAmCherry puncta were counted using the software cell counter tool. For figure 4D, showing an example of how the �-synuclein aggregates (of different sizes) were quantified. The answer to the question: that is effectively what we counted; all aggregates found in the head region of the worm. The aggregates that we can identify have different shapes and sizes, and some that we count as one aggregate could be 2 aggregates that are very close together.

4. Reviewer #1 indicates that in Fig. 1E there is fluorescent dirt, possibly dye remnants on the mCherry channel. We have included a confocal microscope image of another cleaner preparation clearly showing the separation of mCherry (CH1; excitation: 543 nm; emission: 572 ± 20 nm) and SiR700-lysosome channels (CH2; excitation: 635 nm; emission: 655-755). In this picture it is clearly shown that only a fraction of lysosomes is associated with the KFERQ-tagged fluorescent protein. First, myo-3p::KFERQ-PA-mCherry is expressed only in body wall muscle cells and not in pharyngeal muscle and nerve cells. Therefore, as observed in the new Fig 1E, lysosomes stained with the SiR700-lysosome probe in pharyngeal muscle and nerve cells do not correlate with mCherry signal (see Fig. 1E (d, h) ICA (intensity correlation analysis) image). Contrarily, a high intensity correlation between both channels is observed locally in most lysosomes in body wall muscle cells in the head area (see Fig. 1E (c, d, g, h) arrows) with some exceptions (Fig. 1E (c, d, g, h) arrowheads). Colocalization analysis was made locally in body wall muscle cells using the Intensity correlation analysis colocalization method, as shown in Supporting information S1 and described in more detail in materials and methods. The product of the differences from the mean (PDM) of each pixel intensity from each channel was spatially represented in pseudo colored ICA images with yellow indicating high positive values (high correlation) and blue indicating negative values (negative correlation or exclusion).

The merged and ICA images in Fig. 1E clearly show a high degree of colocalization only in lysosomes. Moreover, the absence of colocalization between the diffuse cytosolic PA-mCherry signal and the signal corresponding to the SiR700-lysosome fluorescence channel, suggests no signal bleed-through between the two channels. To corroborate the above statement, the fluorescence bleed-through controls between the different channels are shown. Supporting information Figure S1 now includes fluorescence channel crosstalk controls of worms expressing myo-3p::KFERQ-PA-mCherry and stained with the SiR700-lysosome probe. As shown in Fig S1A no fluorescence bleed-through of the SiR700-Lysosome probe in the mCherry channel (CH1; excitation: 543 nm; emission: 572 ± 20 nm) was detected in non-photoactivated worms. This is more evident in the CH1 image with enhanced contrast by histogram equalization (CH1 Equalized), where no spatial correlation of the fluorescence between both channels is observed. Since the SiR700 in vivo probe enters the worms through ingestion, there is a very strong fluorescence signal in the pharynx cavity. However, nor in this case neither in the intracellular punctate structures of head muscle cells observed in CH2 (CH2; excitation: 635 nm; emission: 655-755) are observed in the mCherry channel 1 (CH1). Similarly, in photoactivated myo-3p::KFERQ-PA-mCherry worms there is no substantial bleed-through of PA-mCherry fluorescence in SiR700-Lysosome fluorescence channel (CH2) as observed in Figure S1B.

5. Reviewer #1 indicates that most of the lysosomal marker is not absorbed in the PA-mCherry-expressing muscles cells of the head and that the two signals somehow anticorrelate at the whole cell level. In all our experiments we observed that the SiR700-Lysosome probe was properly absorbed in body wall muscle cells, however the number of lysosomes and the intensity of the lysosomal marker in pharyngeal cells was greater than in body wall muscle cells. In Fig 1E, we show that the SiR-700-lysosome probe was properly absorbed in KFERQ-PA-mCherry expressing cells. We provided single and merged fluorescent images in full frame figures and zoomed areas as required by reviewer #1.

S2 Fig. (A) The lysosomes are present in all tissues and therefore we observe green dots everywhere. The cherry marker (not KFERQ-PAmCherry) is expressed only in body wall muscle cells. This is a control experiment to show that that cherry and lysosomes (green) do not colocalize, while KFERQ-PAmCherry and lysosomes do. We explained how the colocalization studies were carried out in S1 and its figure legend.

6. Lines 302-304. There is a comment about this in the discussion line 444-456. Studies in human cells have previously demonstrated the existence of an interaction between macroautophagy and CMA [Kausin et al 20218], whereby cells respond to blockade of one of these pathways by up-regulating the other.

7. The goal of Fig. 4B is only to show that the expression of a-synuclein in body wall muscle cells affect the worms’ swimming behavior. Graph 4C compares the swimming behavior of the transgenic worms that express a-synuclein under feeding and starving conditions (when CMA is activated). It has been shown that starvation in WT worms decreases the rate of locomotion (Genetics, Vol. 216, 837–878 December 2020). Our point is to see the effect on a-synuclein.

8. The muscles in the head area are part of the body wall muscles. �-synuclein is expressed in all these muscle cells, as shown in Fig.4Ad. Fig.4Ac shows that the actin cytoskeleton of the body wall muscles is very disorganized. We perform the quantification of a-synuclein aggregates in the head region because it is a well-defined anatomic area.

9. True. We have added statistical tests in each figure legend.

Minor comments:

10. The reviewer is right. Line 69. “Specialized autophagy receptors” was changed to “Selective autophagy receptors”.

11. The word “triggers” was changed to “increases”.

12. Figure 1E was changed

13. Figures S1 was changed to

14. The word “dot” was changed for “puncta” throughout the text.

15. The mistake in line 140 was corrected.

16. line 175 syntaxis changed, protein-protein docking. "The lowest-energy conformer of the most populated cluster of each complex, which also had to be compatible with the steric hindrance of the membrane, was subjected to a 100 ns MD simulation using the same methods described for LMP-1 and LMP-2. RMSD and RMSF of HSP-1 in each simulation were later calculated using MDAnalysis."

17. line 186 We eliminated the workd “embryo”. The word “feed” was replaced with “fed”.

18. We did our preliminary experiments of photoactivation for 10 minutes. However, it was important to limit UV exposure, and we found that 5 minutes was enough to visualize KFERQ-PAmCherry. All experiments shown were done with 5-minute exposure time.

19. Line 210. The term “transfected” was changed for “transformed”.

20. Fig 1D legend. We added 1 asterisk for the p value p 0,0001.

21. Fig S1 and S2 were changed for provide a better explanation

22. The size points in graph Fig. 2B. were adjusted to the same size. There is statistically significant difference between starved control (RNAi) and starved hsp-1 (RNAi): ****p< 0.0001. This was added in the graph Fig 2 B and F

23. We changed axes labelling for consistency in all figures.

24. There was a mistake. We fixed the figure with the appropriate pictures. Thank you very much!

25. We changed the graph to better show that there is in fact a statistical difference in the puncta number per head between starved lmp-1 mutant and starved downregulated hsp-1 in the lmp-1 mutant background. Since hsp-1 is involved in CMA and in microautophagy, it is possible that there is increase macroautophagy due to the crosstalk between these pathways. (Kaushik S and Cuervo AM, 2018)

26. The conditions are different. For example, in Fig 2B the time after photoactivation is 4h until data collection. In Fig 3B, the time after photoactivation is only 2 h until data collection because we are looking at macroautophagy. For each experiment we monitor the same population at different times (of course different individuals from the same plates at different times), therefore there is internal consistency of the results.

27. We changed LC3 for “GABARAP”.

28. We added statistics in the figure legend.

29. Since there is no decrease in the number of puncta when hsp-1 is knockdowned in the lmp-1 mutant background (they are not additive), it means that they could be in the same pathway. Why is there a modest increment in the number of puncta? We could speculate that in the absence of both gene products there could be a compensatory macroautophagy mechanism as seen in mammals (Kaushik S and Cuervo AM. 2018. Mol. Cell Biol. 19, 365-381). It is also important to consider that this gene silencing strategy may not completely suppress the expression of hsp-1. According to this we modified the paragraph in the result section (now 297-301)

30. The reference was included in the text

31. We changed it for 40X.

Reviewer #2

The reviewer is right. What was originally submitted was a draft version of the figures. Therefore, they are of low quality. We have now uploaded the original eps figures, separated from the manuscript, where the puncta can be visualized. It is also important to consider that KFERQ-PAmCherry can always be observed in the cytosol. When the worms are starved, some of this marker is accumulated into the lysosomes. However, since KFERQ-PAmCherry is overexpressed in the muscle cells, much remains in the cytosol.

In the absence of LMP-1 (the KFERQ receptor), the number of puncta in the fed and staved conditions is similar, the is no change, as is the case in the WT (control RNAi). We see puncta under all conditions, including the fed condition. However, only when the receptor (LMP-1) is present we observed increased number of punta.

Figure 2E shows that WT and mutant worms have the same number of active lysosomes (SiR700-lysosome is based on the cathepsin D binding natural product pepstatin A). It is a control to show that the absence of LMP-1 does not affect lysosomal biogenesis and functionality.

---

## [Decision Letter · Decision Letter 1]

6 Apr 2025

PONE-D-24-53431R1KFERQ-selective protein autophagy in Caenorhabditis elegans depends on LMP-1PLOS ONE

Dear Dr. Aldunate,

Thank you for submitting your manuscript to PLOS ONE. After careful consideration, we feel that it has merit but does not fully meet PLOS ONE’s publication criteria as it currently stands. Therefore, we invite you to submit a revised version of the manuscript that addresses the points raised during the review process.

We look forward to receiving your revised manuscript.

Kind regards,

David Chau

Academic Editor

PLOS ONE

Reviewers' comments:

Reviewer's Responses to Questions

**Comments to the Author**

1. If the authors have adequately addressed your comments raised in a previous round of review and you feel that this manuscript is now acceptable for publication, you may indicate that here to bypass the “Comments to the Author” section, enter your conflict of interest statement in the “Confidential to Editor” section, and submit your "Accept" recommendation.

Reviewer #1: (No Response)

Reviewer #2: (No Response)

2. Is the manuscript technically sound, and do the data support the conclusions?

Reviewer #1: Partly

Reviewer #2: Partly

3. Has the statistical analysis been performed appropriately and rigorously? 

Reviewer #1: I Don't Know

Reviewer #2: Yes

4. Have the authors made all data underlying the findings in their manuscript fully available?

Reviewer #1: No

Reviewer #2: No

5. Is the manuscript presented in an intelligible fashion and written in standard English?

Reviewer #1: Yes

Reviewer #2: Yes

6. Review Comments to the Author

Reviewer #1: The authors have responded to all reviewers’ comments and have defended several of them. However, some issues still remain including major ones.

Major

1. The quality of the microscopic images remains low. I do not have access to .eps files and cannot judge the content.

2. The issue concerning puncta quantification is not only to describe the analysis procedure. To my eyes some images cannot be easily subjected to quantification. The masks of quantified particles can answer the above.

3. Indeed, the head muscles are part of the body wall musculature. However, the comment was referring to the thrashing assay and the impact of the rest of the body wall muscle cells in the thrashing assay. Those cells outnumber the head muscle cells and they are at least equally well-defined anatomically.

Minor

4. My title suggestion referred to the grammar. A “destination” cannot be “triggered” or “increased”.

5. The word “dots” still remains in some cases.

6. Fig. 1D legend: In the graph, there are three and not four asterisks.

7. Sup. Fig. 1B and Sup. Fig. 2Bb: Heatmaps are still missing.

8. Reference 59 appears before reference 54.

Reviewer #2: The authors have revised the manuscript.

First a general comment: The authors should provide the response to the reviewers beneath the respective questions. This will make the reviewing easier.

Specifically, the authors have provided images in which the puncta/aggregates could be seen. This was the major concern as the whole study was based on this precinct. Now, the manuscript is a lot better. But, the following concerns need to be addressed.

1. In figure 1C-d, KFERQ- mCherry puncta upon starvation could be seen. The lysosomal targeting image is provided. More than the number of puncta (Fig. 1D), the number of puncta localized to the lysosome could be a better indicator of chaperone induced autophagy (CIA). This needs to be included. Further, this will reduce the reliance on the subjective counting of aggregates with varying sizes and shapes that the authors have provided as an explanation to Reviewer 1’s query.

2. In Fig. 3, lgg-1 RNAi results in increase in puncta number. Does this indicate activation of CIA? A double RNAi for lgg-1 and lmp-1 can clarify this to state that a GABARAP- independent autophagy is contributing to this. This will add strength to lmp-1 role in CIA in C. elegans.

Addition of this data will be ideal. Otherwise, include the potential additional role of lmp-1 in sentence 311.

3. In, Fig.4, alpha-synuclein aggregates increase in lmp-1 RNAi and the aggregate number reduce upon starvation which is the opposite of what is seen with KFERQ-PAmCherry(Fig.1) and lmp-1RNAi/mutant(Fig.2). Hence, is there any involvement of lmp-1 mediated CIA? Does LMP-1 reduce alpha- synuclein by any other mechanism? A lysosomal co-staining would have helped here to show lmp-1 involvement in autophagy and lysosomal targeting or another mechanism.

4. Both Fig.3 and Fig.4, lgg-1 RNAi increasing KFERQ-PAmCherry puncta (Fig.3), and lmp-1 RNAi increasing alpha-synuclein aggregates(fig.4) indicate an overlap and/or some other function of lmp-1. The authors can include this in the discussion.

5. In Discussion sentence 444, for the usage of “mature lysosome” is there any evidence provided? Otherwise, change it to “lysosomes”.

7. PLOS authors have the option to publish the peer review history of their article (what does this mean? ). If published, this will include your full peer review and any attached files.

**Do you want your identity to be public for this peer review?** For information about this choice, including consent withdrawal, please see our Privacy Policy .

Reviewer #1: No

Reviewer #2: No

---

## [Author Response · Author response to Decision Letter 2]

21 May 2025

We understand that the main problem is that high-resolution images are not available to the reviewers in eps or tiff format, and therefore they are of very low quality. The journal process for submission only allows figures to be uploaded with a maximum of 10 Mb, but we were able to upload them in a quality sufficient enough for their analyses. However, when the final version is converted into pdf in the journal’s platform, the quality of the image’s decreases, so we agree with the reviewers that they are difficult to visualize. They must actively search for them in the repository where we include graph data and images in EPS and TIFF format as has been requested (Harvard dataverse “ https://doi.org/10.7910/DVN/DFPZQX”. The reviewers can download all images as well as raw graph data.

3. Has the statistical analysis been performed appropriately and rigorously?

Reviewer #1: I Don't Know

Reviewer #2: Yes

The numerical data (graphs and raw data) and statistical analysis have been uploaded into Harvard database repository https://doi.org/10.7910/DVN/DFPZQX

Moreover, in every figure legend we specified the statistical tests used, the sample size, and the number of independent assays.

4. Have the authors made all data underlying the findings in their manuscript fully available?

The requires authors to make all data underlying the findings described in their manuscript fully available without restriction, with rare exception (please refer to the Data Availability Statement in the manuscript PDF file). The data should be provided as part of the manuscript or its supporting information, or deposited to a public repository. For example, in addition to summary statistics, the data points behind means, medians and variance measures should be available. If there are restrictions on publicly sharing data—e.g. participant privacy or use of data from a third party—those must be specified.

Reviewer #1: No

Reviewer #2: No

All numerical data (graphs and raw data) has been uploaded into Harvard database repository https://doi.org/10.7910/DVN/DFPZQX This link was already available in the answer to the reviewers (point 3) and in the pdf created in the journal´s platform and in “data availability”.

5. Is the manuscript presented in an intelligible fashion and written in standard English?

Reviewer #1: Yes

Reviewer #2: Yes

6. Review Comments to the Author

Reviewer #1: The authors have responded to all reviewers’ comments and have defended several of them. However, some issues remain including major ones.

Major

1. The quality of the microscopic images remains low. I do not have access to .eps files and cannot judge the content.

We understand that the main problem is that high-resolution images are not available to the reviewers in eps or tiff format, and therefore they are of very low quality. The journal process for submission only allows figures to be uploaded with a maximum of 10 Mb, but we were able to upload them in a quality sufficient for their analyses. However, when the final version is converted into pdf in the journal’s platform, the quality of the images decreases in such a way, so we agree with the reviewers that they are difficult to visualize. Unfortunately, they must actively search for them in the repository where we include graph data and images in EPS and TIFF format as has been requested (Harvard dataverse https://doi.org/10.7910/DVN/DFPZQX). All data can be downloaded directly from this platform.

2. The issue concerning puncta quantification is not only to describe the analysis procedure. To my eyes some images cannot be easily subjected to quantification. The masks of quantified particles can answer the above.

We expect that this observation would be answered satisfactorily if the images could be seen in higher quality. Despite this, we included a supplementary (S1) figure with the masks of quantified particles of a set of representative pictures from Fig1C .

It is important to be aware that the expression of KFERQ-PAmCherry is mainly cytosolic and that upon starvation it acquires a punctate distribution. However, some cytosolic fluorescence always remains.

3. Indeed, the head muscles are part of the body wall musculature. However, the comment was referring to the thrashing assay and the impact of the rest of the body wall muscle cells in the thrashing assay. Those cells outnumber the head muscle cells and they are at least equally well-defined anatomically.

The body wall muscles are one tissue, from head to tail. What happens in the head muscles also occurs in the muscles of the rest of the body. Thrashing is an assay that is used to evaluate mobility, which is affected when there is muscle or neuronal defects. In particular, the effects of alpha-synuclein aggregates in muscle cells of this strain have been characterized using this assay. (doi: 10.3390/ph15050512). The autofluorescence of the intestine and presence of eggs in the uterus make the visualization of alpha-synuclein aggregates in body wall muscle cells very difficult and unreliable, except for the head region.

Minor

4. My title suggestion referred to the grammar. A “destination” cannot be “triggered” or “increased”.

The reviewer is correct. We have changed the sentence to: “Starvation promotes KFERQ-PAmCherry delivery to lysosomes in C. elegans muscle cells”.

5. The word “dots” still remains in some cases.

The remaining “dots” words have been changed.

6. Fig. 1D legend: In the graph, there are three and not four asterisks.

The missing asterisk has been added in the graph.

7. Sup. Fig. 1B and Sup. Fig. 2Bb: Heatmaps are still missing.

Heatmaps were added. Sup. Fig 2Bb is now S3Bb.

8. Reference 59 appears before reference 54.

Reference #59 was corrected and incorporated in line 451 (new manuscript).

Reviewer #2: The authors have revised the manuscript.

First a general comment: The authors should provide the response to the reviewers beneath the respective questions. This will make the reviewing easier.

Specifically, the authors have provided images in which the puncta/aggregates could be seen. This was the major concern as the whole study was based on this precinct. Now, the manuscript is a lot better. But the following concerns need to be addressed.

1. In figure 1C-d, KFERQ- mCherry puncta upon starvation could be seen. The lysosomal targeting image is provided. More than the number of puncta (Fig. 1D), the number of puncta localized to the lysosome could be a better indicator of chaperone induced autophagy (CIA). This needs to be included. Further, this will reduce the reliance on the subjective counting of aggregates with varying sizes and shapes that the authors have provided as an explanation to Reviewer 1’s query.

The results in Figure 1 E show that almost 100% of the KFERQ-PAmCherry puncta co-localize with the lysosomal marker in muscle cells. The confocal images in Fig 1E also show that the synchronization of the lysosomal signal (Sir700-lysosome) increases with that of the positive KFERQ-PAmCherry particles, occurring in all cases (Fig E, e-h). However, there are few lysosomal particles that do not co-locate with KFERQ (Fig E, f-g, arrowhead), either because they are not positive for CMA or because they are found in muscle-adjacent tissues (pharynx, hypodermis, and neurons) that do not express the fluorescent reporter.

Besides the analysis we performed of image in Fig. 1E, we analyzed several other confocal images finding the same results (images below). First, we identified the KFERQ-PAmCherry expressing tissue (red), then we counted the puncta. We found that in contrast to the results with PAmCherry overexpression in Supplementary Fig S3, almost 100% KFERQ-PAmCherry puncta colocalized with Sir700-lysosome in head muscle cells, with only a few puncta of smaller size per image being exclusively Sir700-lysosome positive (and we cannot be sure they are in other tissues). All the other green puncta are in tissues that do not express KFERQ-PAmCherry and therefore are not body wall muscle tissue.

See file response to reviewrs

Fig 1. Confocal microscopy images of C. elegans head muscles expressing the CMA reporter, in which colocalization analysis was performed between A) PA-mCherry (deleted KFERQ) and Sir700-lysosome and B) KFERQ-PA-mCherry- and Sir700-lysosome marker (ICA, corresponding to Supplementary Figure 1C and S3).The total number of fluorescent spots in the muscle region that showed colocalization, the number of spots labeled with the lysosomal tracer alone and the fluorescent spots in the PA-mCherry channel were quantified. The results indicated that only 20% of lysosomes present in muscle cells did not colocalize with the CMA marker (arrowheads). In contrast, less than 0.1 % of the total number of fluorescents on average corresponded to KFERQ- PA-mcherry- that did not colocalize with the lysosomal activity marker (Fig 1B, arrows). (Scale bar = 5 µm)

2. In Fig. 3, lgg-1 RNAi results in increase in puncta number. Does this indicate activation of CIA? A double RNAi for lgg-1 and lmp-1 can clarify this to state that a GABARAP- independent autophagy is contributing to this. This will add strength to lmp-1 role in CIA in C. elegans.

Addition of this data will be ideal. Otherwise, include the potential additional role of lmp-1 in sentence 311.

We thank the reviewer for the insightful and constructive comments, which have helped improve the clarity and quality of our manuscript. The reviewer is right. It is possible that the phenotype of the double RNAi for lgg-1 and lmp-1 could add strength to lmp-1 role in CIA in C. elegans.

To address this, as the reviewer suggestion w–e included in sentence 311 “Additionally, in the absence of macroautophagy, lmp-1 expression can increase, and its stabilization at the lysosomal membrane could be enhanced to support increased CMA flux [1].”

3. In, Fig.4, alpha-synuclein aggregates increase in lmp-1 RNAi and the aggregate number reduce upon starvation which is the opposite of what is seen with KFERQ-PAmCherry(Fig.1) and lmp-1RNAi/mutant(Fig.2). Hence, is there any involvement of lmp-1 mediated CIA? Does LMP-1 reduce alpha- synuclein by any other mechanism? A lysosomal co-staining would have helped here to show lmp-1 involvement in autophagy and lysosomal targeting or another mechanism.

Fig. 4A does not show that the alpha-synuclein aggregates increase in lmp-1 RNAi. In fact, it shows that the absence of lmp-1 under starvation conditions does not decrease the number of aggregates as it happens when lmp-1 is present. It has been shown that CMA is able to deliver alpha-synuclein monomers to the lysosome, but not the aggregates (J Clin Invest. 2008 Feb;118(2):777-88. doi: 10.1172/JCI32806). Therefore, our evidence suggests that the decrease in the total number of aggregates is a consequence of this process of decreased number of monomers available for aggregation. We do not expect that CMA influences the process of aggregation directly.

4. Both Fig.3 and Fig.4, lgg-1 RNAi increasing KFERQ-PAmCherry puncta (Fig.3), and lmp-1 RNAi increasing alpha-synuclein aggregates(fig.4) indicate an overlap and/or some other function of lmp-1. The authors can include this in the discussion.

The answer to this query is the same as for the reviewer’s comment 3.

5. In Discussion sentence 444, for the usage of “mature lysosome” is there any evidence provided? Otherwise, change it to “lysosomes”.

The word “mature” was deleted.

---

## [Decision Letter · Decision Letter 2]

22 Jun 2025

PONE-D-24-53431R2KFERQ-selective protein autophagy in Caenorhabditis elegans depends on LMP-1PLOS ONE

Dear Dr. Aldunate,

Thank you for submitting your manuscript to PLOS ONE. After careful consideration, we feel that it has merit but does not fully meet PLOS ONE’s publication criteria as it currently stands. Therefore, we invite you to submit a revised version of the manuscript that addresses the points raised during the review process.

We look forward to receiving your revised manuscript.

Kind regards,

David Chau

Academic Editor

PLOS ONE

Journal Requirements:

Reviewers' comments:

Reviewer's Responses to Questions

**Comments to the Author**

1. If the authors have adequately addressed your comments raised in a previous round of review and you feel that this manuscript is now acceptable for publication, you may indicate that here to bypass the “Comments to the Author” section, enter your conflict of interest statement in the “Confidential to Editor” section, and submit your "Accept" recommendation.

Reviewer #1: (No Response)

Reviewer #2: (No Response)

2. Is the manuscript technically sound, and do the data support the conclusions?

Reviewer #1: (No Response)

Reviewer #2: Partly

3. Has the statistical analysis been performed appropriately and rigorously? 

Reviewer #1: (No Response)

Reviewer #2: I Don't Know

4. Have the authors made all data underlying the findings in their manuscript fully available?

Reviewer #1: Yes

Reviewer #2: No

5. Is the manuscript presented in an intelligible fashion and written in standard English?

Reviewer #1: Yes

Reviewer #2: Yes

6. Review Comments to the Author

Reviewer #1: In this round of review authors have responded to all reviewers’ comments. There is now access to high resolution images and the representative masks (Fig. S1) are convincing. Apart from the point below I have no further comments.

Without experimental data, the response “What happens in the head muscles also occurs in the muscles of the rest of the body” is at the best case rather vague and misleading. Intestine is a separate tissue and its autofluorescence has characteristic pattern, it can be discriminated from muscle aggregates and the two tissues do not always align during imaging (both at xy as well as the z axis). The same applies for eggs. Even if they interfere with the smooth imaging of muscles cells, this should not affect the existence of aggregates. Imaging of aggregates in some of these cells that do not interfere with intestine or eggs should be feasible. Quantification would be ideal but at least the respective images are necessary. Also, are the muscle cells in Fig. 4A located somewhere at the midbody (other than the head)? The images look somehow tilted. Please include the merge one and scale bar.

Reviewer #2: This manuscript has some interesting findings. Though it has a huge limitation of quantitation of puncta/aggregates being subjective, it can be accepted due to the lack of alternative means to measure the puncta/aggregates,

7. PLOS authors have the option to publish the peer review history of their article (what does this mean? ). If published, this will include your full peer review and any attached files.

**Do you want your identity to be public for this peer review?** For information about this choice, including consent withdrawal, please see our Privacy Policy .

Reviewer #1: No

Reviewer #2: No

---

## [Author Response · Author response to Decision Letter 3]

18 Jul 2025

Answer to Reviewer

6. Review Comments to the Author

Reviewer #1: In this round of review authors have responded to all reviewers’ comments. There is now access to high resolution images and the representative masks (Fig. S1) are convincing. Apart from the point below I have no further comments.

Without experimental data, the response “What happens in the head muscles also occurs in the muscles of the rest of the body” is at the best case rather vague and misleading. Intestine is a separate tissue and its autofluorescence has characteristic pattern, it can be discriminated from muscle aggregates and the two tissues do not always align during imaging (both at xy as well as the z axis). The same applies for eggs. Even if they interfere with the smooth imaging of muscles cells, this should not affect the existence of aggregates. Imaging of aggregates in some of these cells that do not interfere with intestine or eggs should be feasible. Quantification would be ideal but at least the respective images are necessary.

To answer the reviewer’s concern, we put together a series of pictures showing a-synuclein aggregates in muscle cells corresponding to the head, anterior mid-body, posterior mid-body, and tail region (Figure 1). Panel A, shows control animals and B, animals starved for 6 hours. The photographs on the left column were taken using an epifluorescence microscope adding some transmitted light, to visualize the pharynx in the head region, and other anatomical details in other parts of the body. Aggregates are seen in muscle cells in all regions of the worm’s body (head, anterior mid-body, posterior mid-body, and tail). Fewer aggregates are seen in the worms that were starved for 6 h (Panel B).

It is important to emphasize that our decision to quantify aggregates in the head region of individual worms, rather than in other areas, was based on the fact that this region can be readily and consistently delineated by the anatomical landmark of the pharynx, making quantification more reliable.

Moreover, C. elegans body wall muscle cells are similar to one another, arranged in four longitudinal bands and are all connected through gap junctions (Front Physiol. 2014 Feb 11;5:40. doi: 10.3389/fphys.2014.00040. Gap junctions in C. elegans. Karina T Simonsen 1, Donald G Moerman 2, Christian C Naus 1), from head to tail. All contribute to the worm’s sinusoidal crawling and swimming.

Also, are the muscle cells in Fig. 4A located somewhere at the midbody (other than the head)? The images look somehow tilted. Please include the merge one and scale bar.

The figure 4 A shows approximately the same mid-body region, but from different worms of the same strain. This approach was necessary due to technical incompatibilities between the phalloidin staining and GFP/YFP visualization. For phalloidin F-actin staining, worms must be frozen in liquid nitrogen, lyophilized, and treated with acetone (WormBook and Bio Protoc. 2021 Oct 5;11(19):e4183. doi: 10.21769/BioProtoc.4183). However, GFP fluorescence is abolished by acetone treatment, and GFP is also highly sensitive to dehydration. Therefore, aggregates were visualized in vivo. Scale bars were added.

Figure 1. Fluorescence microscopy images of muscle cells from control animals (A) and starved for 6 hours animals (B)The photographs on the left column were taken using an epifluorescence microscope adding some transmitted light, to visualize the pharynx in the head region, and other anatomical details in other parts of the body. Aggregates are seen in muscle cells in all regions of the worm’s body (head, anterior mid-body, posterior mid-body, and tail). Fewer aggregates are seen in the worms that were starved for 6 h (Panel B).

Reviewer #2: This manuscript has some interesting findings. Though it has a huge limitation of quantitation of puncta/aggregates being subjective, it can be accepted due to the lack of alternative means to measure the puncta/aggregates.

---

## [Decision Letter · Decision Letter 3]

31 Jul 2025

KFERQ-selective protein autophagy in Caenorhabditis elegans depends on LMP-1

PONE-D-24-53431R3

Dear Dr. Aldunate,

We’re pleased to inform you that your manuscript has been judged scientifically suitable for publication and will be formally accepted for publication once it meets all outstanding technical requirements.

Kind regards,

David Chau

Academic Editor

PLOS ONE

Additional Editor Comments (optional):

Reviewers' comments:

Reviewer's Responses to Questions

**Comments to the Author**

1. If the authors have adequately addressed your comments raised in a previous round of review and you feel that this manuscript is now acceptable for publication, you may indicate that here to bypass the “Comments to the Author” section, enter your conflict of interest statement in the “Confidential to Editor” section, and submit your "Accept" recommendation.

Reviewer #1: All comments have been addressed

2. Is the manuscript technically sound, and do the data support the conclusions?

Reviewer #1: Partly

3. Has the statistical analysis been performed appropriately and rigorously? 

Reviewer #1: Yes

4. Have the authors made all data underlying the findings in their manuscript fully available?

Reviewer #1: Yes

5. Is the manuscript presented in an intelligible fashion and written in standard English?

Reviewer #1: Yes

6. Review Comments to the Author

Reviewer #1: (No Response)

7. PLOS authors have the option to publish the peer review history of their article (what does this mean? ). If published, this will include your full peer review and any attached files.

**Do you want your identity to be public for this peer review?** For information about this choice, including consent withdrawal, please see our Privacy Policy .

Reviewer #1: No

---

## [Editor Report · Acceptance letter]

PONE-D-24-53431R3

PLOS ONE

Dear Dr. Aldunate,

I'm pleased to inform you that your manuscript has been deemed suitable for publication in PLOS ONE. Congratulations! Your manuscript is now being handed over to our production team.

Kind regards,

on behalf of

Dr. David Chau

Academic Editor

PLOS ONE